# Chromatin protein complexes involved in gene repression in lamina-associated domains

Stefano G Manzo [1,2,3], Abdelghani Mazouzi [2,4], Christ Leemans[1,2], Tom van Schaik[1,2], Nadia Neyazi[1,2], Marjon S van Ruiten[5], Benjamin D Rowland [5], Thijn R Brummelkamp[2,4] & Bas van Steensel [1,2,6✉]

## Abstract

Lamina-associated domains (LADs) are large chromatin regions that are associated with the nuclear lamina (NL) and form a repressive environment for transcription. The molecular players that mediate gene repression in LADs are currently unknown. Here, we performed FACS-based whole-genome genetic screens in human cells using LAD-integrated fluorescent reporters to identify such regulators. Surprisingly, the screen identified very few NL proteins, but revealed roles for dozens of known chromatin regulators. Among these are the negative elongation factor (NELF) complex and interacting factors involved in RNA polymerase pausing, suggesting that regulation of transcription elongation is a mechanism to repress transcription in LADs. Furthermore, the chromatin remodeler complex BAF and the activation complex Mediator can work both as activators and repressors in LADs, depending on the local context and possibly by rewiring heterochromatin. Our data indicate that the fundamental regulators of transcription and chromatin remodeling, rather than interaction with NL proteins, play a major role in transcription regulation within LADs.

**Keywords** Lamina-associated Domains; Gene Repression; Chromatin; Nuclear Lamina; Mediator
**Subject Categories** Chromatin, Transcription & Genomics; Genetics, Gene Therapy & Genetic Disease; Methods & Resources

## Introduction

### Definition and characteristics of LADs

In most metazoan cells, about one-third of the genome is packaged into lamina-associated domains (LADs), which are large, often megabase-sized segments that are in close contact with the nuclear lamina (NL) (Guelen et al, 2008). LADs tend to be highly conserved (Meuleman et al, 2013) and are thought to play important roles in the overall spatial organization of the genome (Falk et al, 2019). In addition, LADs have morphological and biochemical features of heterochromatin, are replicated late during S-phase, and overlap strongly with B compartment, as detected by Hi-C technology (Alagna et al, 2023; Briand and Collas, 2020; Guerreiro and Kind, 2019; Manzo et al, 2022; van Steensel and Belmont, 2017).

### LADs are potent repressive domains

Perhaps the most striking feature of LADs is their link to gene repression. Genes embedded in LADs are generally expressed at much lower levels than genes in inter-LAD regions (iLADs) (Guelen et al, 2008; Peric-Hupkes et al, 2010; Pickersgill et al, 2006). Moreover, during cell differentiation, genes are typically downregulated when they move towards the NL, and often become upregulated when they detach from the NL (Ahanger et al, 2021; Madsen-Østerbye et al, 2022; Peric-Hupkes et al, 2010; Robson et al, 2016). For some genes, it has been observed that detachment from the NL precedes upregulation (Isoda et al, 2017; Peric-Hupkes et al, 2010), suggesting that detachment is a prerequisite for gene activation. Conversely, during *D. melanogaster* neuronal development, the *hunchback* gene becomes irreversibly repressed when it moves towards the NL, but not when this relocation is prevented (Kohwi et al, 2013). Importantly, when active promoters are inserted in LADs, their activity can be up to ~100-fold lower than in inter-LAD regions (Akhtar et al, 2013; Leemans et al, 2019). Furthermore, lowly active human promoters that are naturally located in LADs tend to become more active when they are taken out of their LAD context and placed in an episomal vector in the same cells (Leemans et al, 2019). Together, these data strongly indicate that LADs are potent repressive domains.

### Possible role of the NL in gene repression

The molecular mechanisms responsible for this repression remain largely unknown. One possibility is that physical contacts with the NL play a role. Indeed, artificial tethering of a chromosomal locus to the NL can lead to reduced expression of a subset of genes around the tethered locus (Finlan et al, 2008; Kumaran and Spector,

[1]Division of Gene Regulation, Netherlands Cancer Institute, Amsterdam, the Netherlands. [2]Oncode Institute, Amsterdam, the Netherlands. [3]Department of Biosciences, Università degli Studi di Milano, Via Celoria 26, 20133 Milan, Italy. [4]Division of Biochemistry, Netherlands Cancer Institute, Amsterdam, the Netherlands. [5]Division of Cell Biology, Netherlands Cancer Institute, Amsterdam, the Netherlands. [6]Division of Molecular Genetics, Netherlands Cancer Institute, Amsterdam, the Netherlands. ✉E-mail: b.v.steensel@nki.nl

2008; Reddy et al, 2008). In addition, single-cell analysis of naturally fluctuating NL contacts indicates that genes in LADs are somewhat more highly expressed when stochastically detached from the NL (Rooijers et al, 2019). However, other evidence questions a direct repressive role of NL contacts. For example, during oncogene-induced senescence many genes move towards the NL, but they are not repressed (Lenain et al, 2017). Furthermore, in *C. elegans* Cec-4 mediates genome-wide tethering of heterochromatic regions to the NL, but its deletion caused upregulation of only a single gene (Gonzalez-Sandoval et al, 2015), suggesting that NL contacts are for most genes in LADs not essential for repression.

## Search for NL proteins mediating repression

For a better understanding of the role of NL contacts in gene regulation, it is necessary to identify NL-associated proteins that mediate transcriptional repression. So far, efforts towards this goal have provided only a fragmentary picture. In *C. elegans*, mutation of several NL proteins leads to the derepression of promoters in artificial heterochromatin arrays (Mattout et al, 2011). Both in *D. melanogaster* and mammalian cells, depletion of lamins (the main structural proteins of the NL) causes deregulation of dozens to hundreds of genes—but not preferentially in LADs, suggesting that indirect effects may dominate the observed changes in gene activity (Amendola and van Steensel, 2015; Nazer et al, 2018; Ulianov et al, 2019; Zheng et al, 2018). One laminopathy-associated mutation in Lamin A causes upregulation and NL detachment of some genes within LADs in human cardiomyocytes (Shah et al, 2021). Deletion of human Lamin B Receptor (LBR), another transmembrane protein of the NE, causes stochastic upregulation of the LAD-embedded gene *ABCB1*, but only in a very small minority of cells (Manjon et al, 2023). Furthermore, three transmembrane proteins of the nuclear envelope (NE) were found to control both the peripheral positioning and expression of genes in the human myogenic cell lineage and thus may be direct regulators in this lineage (Robson et al, 2016). Several other studies have reported that altered levels or mutations of certain NL proteins cause deregulation of gene expression (Briand et al, 2018; Oldenburg et al, 2014; Solovei et al, 2013); however, lack of NL interaction maps in these studies has made it difficult to discern direct from indirect effects. Thus, evidence for roles of NL proteins in the repression of LAD genes is still anecdotal.

## Limited role of H3K9 methylation

In addition to the putative repressive role of NL proteins, involvement of proteins that are part of LAD chromatin should be considered. LADs are frequently marked by the heterochromatic histone modifications H3K9me2 and H3K9me3 (Guelen et al, 2008; Wen et al, 2009). In *C. elegans* these marks appear to play a role, at least in the repression of NL-associated heterochromatic transgene arrays (Towbin et al, 2012). To our knowledge, genome-wide effects of depletion of H3K9me2 or H3K9me3 on gene expression in mammalian LADs have not been reported. The H3K9 methyltransferases G9a/GLP and SETDB1 were found to repress transcription of some genes located in the B compartment, which tends to overlap with LADs (Fukuda et al, 2021). However, even in mouse cells in which all six H3K9 methyltransferases

were depleted, fewer than 20% of all genes marked by either H3K9me2 or H3K9me3 were upregulated, even though all heterochromatin (including at the NL) was morphologically lost (Montavon et al, 2021). Assuming that the majority of genes marked by H3K9me2 and H3K9me3 are located in LADs, these data suggest that these histone marks explain only part of gene repression in LADs.

## Other candidate chromatin-mediated mechanisms

It thus seems that other features of LAD chromatin may be involved. Accessibility of enhancers and promoters to certain transcription factors may be limited in LADs (Yao et al, 2011), and this exclusion may be controlled by as yet unknown proteins. Furthermore, genes in LADs exhibit frequent promoter-proximal stalling of RNA polymerase II (PolII) (Leemans et al, 2019; Wu and Yao, 2017). This suggests that transcription elongation is impaired in LADs, but the underlying mechanism has remained unexplained.

## Some genes escape repression in LADs

It should be noted that ~10% of all genes in LADs retain relatively high levels of transcription (Ahanger et al, 2021; Leemans et al, 2019). Analysis of such "escaper" genes has suggested that local chromatin context, local strength of NL contacts and promoter architecture are features that may modulate the sensitivity of promoters to the LAD environment (Leemans et al, 2019; Madsen-Østerbye et al, 2022). However, proteins that may facilitate the activation of genes inside LADs have not been identified.

## Scope of the study and summary of the results

Here, we took an unbiased approach to identify LAD-specific transcriptional regulators in human cells. We designed fluorescence-activated cell sorting (FACS)-based genome-wide haploid genetic screens in HAP-1 cells to systematically identify proteins that control the expression of a repressed reporter gene that was inserted in two different LADs. Remarkably, the screen did not yield evidence for repressive roles of NL proteins, but instead identified a variety of chromatin complexes as regulators of gene expression in LADs. Among these are regulators of PolII pausing and transcription elongation, the chromatin remodeler BAF, and the transcriptional regulator Mediator. The latter two can both activate and repress genes in LADs, indicating that their role depends strongly on the local context.

# Results

## Haploid genetic screens to identify key regulators of gene activity in LADs

### Overview of the screens

To identify transcriptional regulators in LADs we performed FACS-based genetic screens in human cells. We engineered haploid HAP1 cells to contain a reporter system designed to monitor gene expression within LADs (Fig. 1A). We then conducted insertional mutagenesis screens coupled with FACS-based sorting using the engineered reporters as molecular readout.

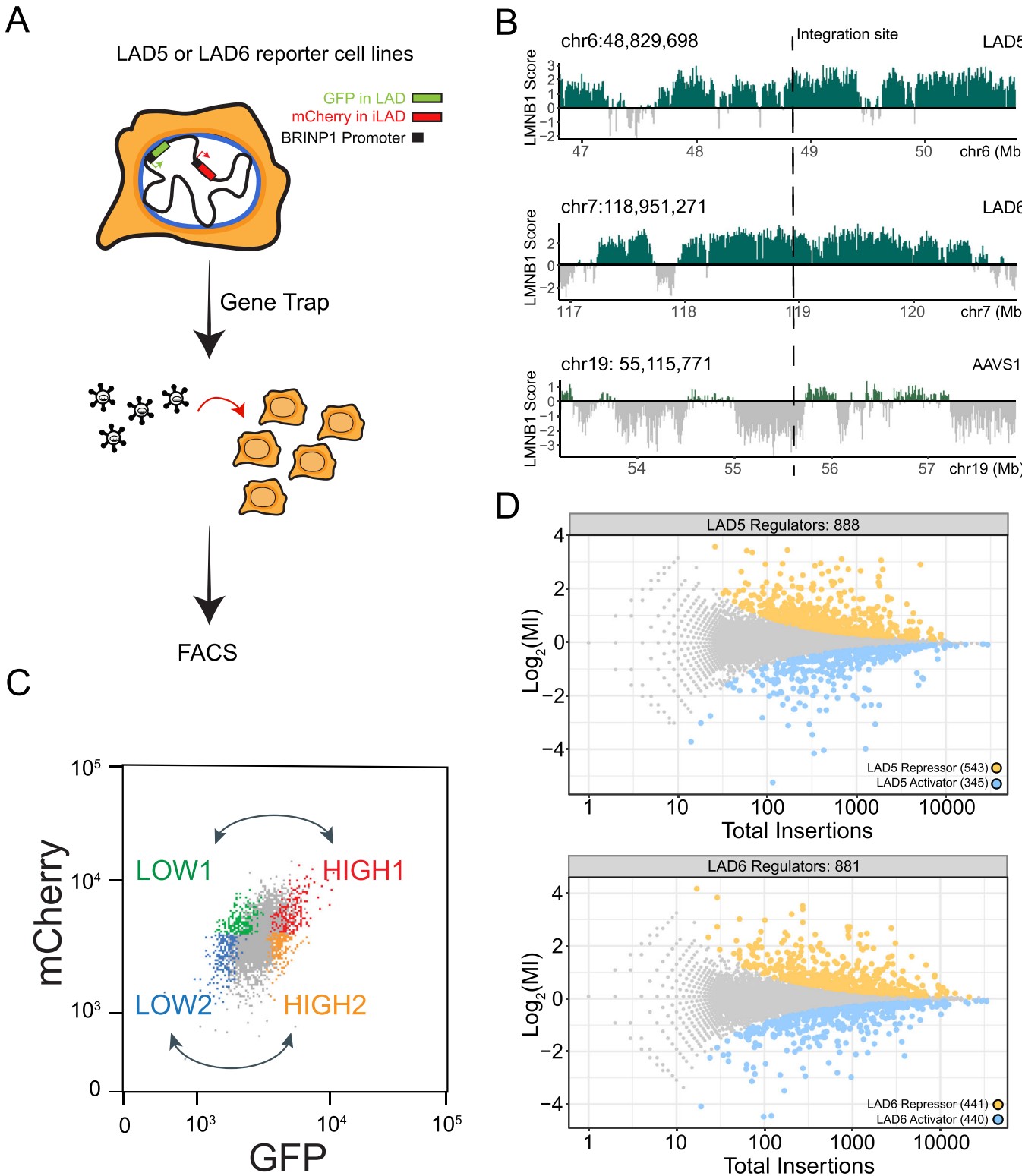

### Selection of reporter promoter

To probe LAD-mediated gene regulation, we constructed a GFP expression cassette under the control of the *BRINP1* promoter (*pBRINP1::GFP*). Previous analysis demonstrated that this promoter is sensitive to repression when located inside LADs in K562 cells

(Leemans et al, 2019). Similarly, in HAP1 cells the *BRINP1* gene is inside a LAD and expressed at very low levels (Fig. EV1A); however, when *pBRINP1::GFP* is transiently transfected as a plasmid (i.e., not integrated in a LAD) it is highly expressed (Fig. EV1B). These data indicate that *pBRINP1* in HAP1 cells has

**Figure 1.   Designing a screen for LAD-specific transcriptional regulators.**

(A) Overview of the haploid genetic screen design. The *pBRINP1::GFP* reporter was integrated in two different LADs generating two clonal HAP1 cell lines. In the same cell lines *pBRINP1::mCherry* was integrated in the inter-LAD locus *AAVS1*. Cells were transduced with a gene-trap lentivirus to generate random knockouts and sorted according to fluorescence intensity. (B) Genomic tracks of LMNB1 DamID signal showing the two LADs (LAD5 and LAD6) and the *AAVS1* locus selected as integration sites. (C) Sorting strategy for comparison of HIGH and LOW GFP in gene trapped cells. (D) Fishtail plots showing the mutational index (y-axis, MI) and total number of insertions (x-axis) for all genes. Non-significant genes are colored in gray, significant hits (fcpv < 0.05) are colored in yellow for LAD reporter repressors (positive MI) and in blue for LAD reporter activators (negative MI). Top panel: results for LAD5. Bottom panel: results for LAD6. Screens results are from one biological replicate.

the intrinsic ability to drive transcription, but it is repressed by the LAD environment.

### Two dual-color reporter cell lines

We then inserted the *pBRINP1::GFP* reporter into two different LADs (here named LAD5 and LAD6; Fig. 1B) by conventional Cas9 editing, resulting in two separate clonal cell lines. We chose two different LADs in anticipation of possible LAD-specific mechanisms of gene regulation. Analysis of five histone modifications revealed that the two LAD reporters were embedded in substantially different chromatin environments despite quite similar interactions with the NL. The LAD5 reporter was located in a chromatin type typical of weak transcription and in close proximity to a cluster of putative regulatory elements. Instead, the LAD6 reporter was located in chromatin lacking any of the tested chromatin marks (Fig. EV1C). In the same cell lines we also integrated a second reporter, *pBRINP1::mCherry*, into the inter-LAD locus *AVVS1* (Fig. 1A). This served as an internal reference to distinguish general regulators of *pBRINP1*, which should equally affect GFP and mCherry expression, from LAD-specific regulators that should affect GFP but not mCherry expression levels.

### Controls for proper integration and expression

Genotyping by PCR and tagmentation mapping confirmed that the reporters were integrated exclusively in the intended locations. Detailed analysis pointed to the presence of tandem integrations of the mCherry reporters in the *AVVS1* locus in both cell lines, and of the GFP reporter in LAD6 (Appendix Fig. S1A–C and Methods). In both cell lines RT-qPCR analysis showed substantially lower levels of mRNA from the GFP reporter compared to the mCherry reporter (Appendix Fig. S1D). Although the unknown exact copy number of the tandem insertions precluded normalization of the expression levels, these data indicated that *pBRINP1* in LADs was more repressed than in the *AVVS1* locus, in agreement with the well-described repressive environment of LADs (Akhtar et al, 2013; Leemans et al, 2019).

### Mutagenesis, cell sorting, and score calculation

We then subjected the two cell lines to insertional mutagenesis with a gene trap retroviral vector that effectively disrupts gene function upon integration (Blomen et al, 2015; Brockmann et al, 2017; Mazouzi et al, 2023) (Fig. 1A). Next, we sorted the cells into two pairs of bins with low and high GFP levels, but with similar mCherry intensity (Fig. 1C; Appendix Fig. S2A). In each population, we then mapped the insertions as described (Brockmann et al, 2017; Mazouzi et al, 2023). Identification and counting of the number of sense insertions for a given gene in the high and low GFP population yielded a mutational index (MI) reflecting enrichment in either the high or low pool, and an associated FDR-corrected p-value (*fcpv*, see Methods).

### Robustness of screen results

To assess the robustness of these scores, we conducted GFP-LOW versus GFP-HIGH comparisons in both the mCherry-LOW and mCherry-HIGH bin pairs (Fig. 1C). Among the genes with significant scores (*fcpv* < 0.05), the correlation of MI scores was high between these bin pairs (R = 0.85 and 0.80 for LAD5 and LAD6, respectively; Appendix Fig. S2B), indicating that the results are robust and largely independent of intrinsic cell-to-cell variation in gene activity. We therefore merged the data from the LOW and HIGH pairwise comparisons, resulting into a joint MI score (see Methods).

### Interpretation of screen results

A positive MI value of a gene indicates that disruption of the gene caused preferential upregulation of the expression of the LAD-inserted reporter, and hence that the gene encodes a putative LAD-specific repressor. Conversely, a negative MI value indicates that disruption of the gene caused preferential downregulation of the expression of the LAD-inserted reporter, and hence the gene may encode a putative LAD-specific activator. We thus identified 888 putative regulators for LAD5 (543 putative repressors and 345 putative activators, *fcpv* < 0.05) and 881 for LAD6 (441 putative repressors and 440 putative activators, *fcpv* < 0.05) (Fig. 1D). However, it is likely that many of these hits are due to indirect effects. For example, disruption of metabolic pathways or signal transduction molecules on the cell membrane or in the cytoplasm may alter the state of the cells, which could cause changes in NL interactions that in turn affect reporter expression. To increase the chances of identifying direct regulators, we focused our further analyses on genes encoding proteins that were previously annotated as components of the NL, nuclear envelope (NE), or chromatin, or as regulators of transcription.

## Very few NL proteins are identified as LAD-specific repressors

### No enrichment of nuclear envelope proteins

Surprisingly, very few of the hits identified in the screens overlapped with known NE proteins (including known NL proteins) as listed in the Gene Ontology (GO) database (Gene Ontology et al, 2023): only 16 and 15 proteins for LAD5 and LAD6, respectively (Fig. 2A; Appendix Fig. S2C). This is not significantly more than may be expected by random chance (Fig. 2C). Moreover, for several of these proteins it is questionable whether the GO annotation as component of the NE is correct (e.g., FANCL, INSR, SIRT1, and RBM15). Nevertheless, one noteworthy hit among the putative repressors was Lamin B1 (LMNB1), although it was only modestly enriched in the LAD5 screen (MI = 0.44, *fcpv* = 0.00023), and not a significant hit for LAD6 (MI = −0.012, *fcpv* = 1). LMNB1 may thus mildly contribute to gene repression in a subset of LADs.

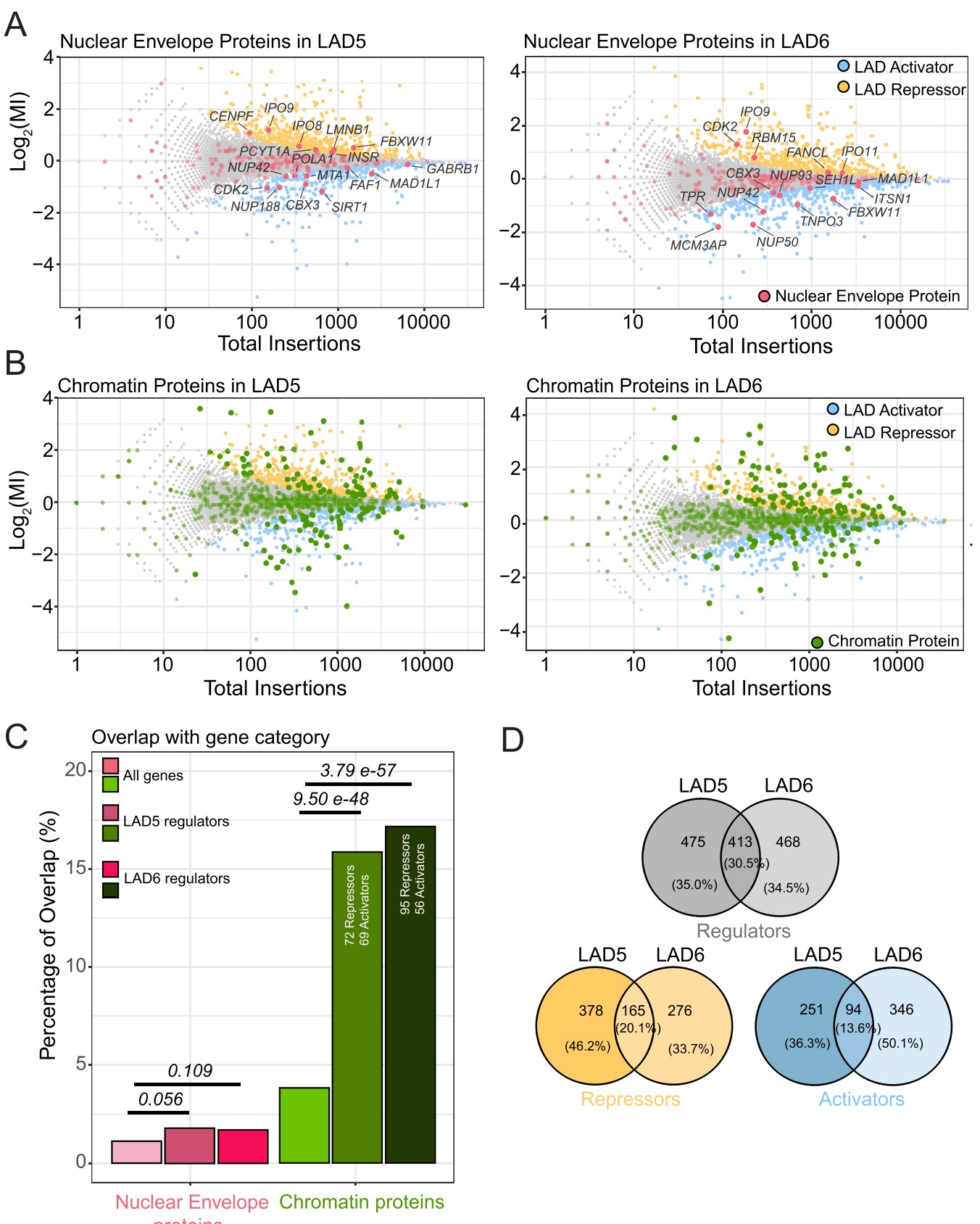

**Figure 2. Chromatin proteins rather than NL components modulate expression of LAD reporters.**

(**A**) Screen hits (LAD activators and repressors) annotated as NE proteins (red, GO category GO:0005635). (**B**) Screen hits annotated as chromatin proteins (green, GO:0006325, GO:0140110, and GO:0003682 combined). (**C**) Enrichment analysis for NE and chromatin proteins in the LAD reporter regulators dataset. Statistical significance was calculated with Fisher's Exact Test. (**D**) Partial overlap of LAD5 and LAD6 regulators (gray Venn diagram for all LAD regulators, yellow for repressors, blue for activators).

### Putative activators are primarily linked to nuclear pores

Among the putative LAD-specific activators (MI < 0) were five proteins of the nuclear pore complex (NUP42, NUP188, NUP50, NUP93 and TPR), one of which (NUP42) is shared between the LAD5 and LAD6 hits (Fig. 2A). Indeed, the nuclear pore complex has been implicated in transcription activation (Sumner and Brickner, 2022), possibly by repelling chromatin from the NL (Boumendil et al, 2019).

### Proteomics data underscores paucity of NE proteins

As the GO annotations are likely to be incomplete, we also queried a list of proteins that were biochemically determined to be enriched in the NE (Cheng et al, 2023). This yielded no additional putative repressors for LAD5, and only two for LAD6 (Fig. EV2A). Of the latter, SENP1 is a nucleoplasmic peptidase that removes SUMO from various transcription regulators (Cui et al, 2017; Gao et al, 2022), while GPAT4 is a cytoplasmic glycerol-3-phosphate acyltransferase (Karasawa et al, 2019). Because immunofluorescence microscopy has not confirmed enrichment of these proteins at the NE (Human Protein Atlas, (Uhlén et al, 2015)) and they were only detected in the LAD6 screen, we decided not to investigate these proteins further. The few putative activators that overlap with biochemically defined NE proteins again included primarily proteins of the nuclear pore complex.

### Gene essentiality is unlikely to explain lack of NL/NE hits

We considered the possibility that some repressors of gene activity in LADs are essential for fitness of HAP1 cells; these would not be detected in our screens. Comparison to previously determined estimates of the impact of each gene on fitness of HAP1 cells (Blomen et al, 2015) indicated that about 25–30% of NL/NE proteins fall in this category (Fig. EV2B). However, the vast majority of these proteins are components of the nuclear pore complex or of the nuclear transport machinery. Considering our results described above for multiple non-essential nuclear pore proteins, it seems unlikely that these proteins mediate repression. Nevertheless, we cannot rule out that other essential NL/NE proteins such as BANF1 are involved in gene repression in LADs. In summary, we failed to obtain evidence for a prominent role of any well-established NL/NE protein in mediating repression in LADs, except for a modest contribution of LMNB1. We cannot, however, exclude that NL/NE proteins act redundantly in this process.

## A broad diversity of chromatin proteins is involved in LAD-specific regulation

### Many hits are chromatin proteins

In contrast to this paucity of NE/NL proteins, the screens identified 141 and 151 proteins, for LAD5 and LAD6 respectively, that are known regulators of transcription and chromatin according to the GO database (Fig. 2B; see figure legend for Gene Ontology terms).

This is about 4-fold more than can be expected by random chance (Fig. 2C). This enrichment underscores the overall specificity of the screen. For LAD5, 72 putative repressors and 69 putative activators are chromatin proteins, while for LAD6 we could identify in this category 95 putative repressors and 56 putative activators. Systematic GO analysis using G:Profiler (Kolberg et al, 2023) also showed enrichment of genes with annotations such as "nucleoplasm", "transcription coregulator activity", and "histone modifying activity", among others (Appendix Fig. S2C). We conclude that control of expression of the reporter genes by LADs is mediated by a large number of chromatin regulators, while the role of NL proteins appears to be very limited.

### Limited overlap of hits between the two LADs

Surprisingly, even though both screens showed similar enrichments of chromatin regulators, the overlap of the hits between the two LADs was rather modest. Only 30.5% of the combined hits for the two LADs were shared between the two datasets, with putative LAD activators and LAD repressors showing about 14% and 20% of overlap between the two LADs, respectively (Fig. 2D). Most likely, this reflects heterogeneity in the local chromatin composition of the two LADs, as described above (Fig. EV1C). Below we will highlight some striking similarities and differences.

## Regulators of RNA polymerase II elongation repress reporter activity in LADs

### Entire NELF complex is a putative repressor in both LADs

Among the putative repressor proteins that were common between LAD5 and LAD6, subunits of NELF (negative elongation factor) were prominent hits (Fig. 3A). All four subunits of this protein complex (named A, B, CD, and E) showed strongly positive and significant MI scores for both LADs, suggesting that the entire complex is required for the repression of both LAD-integrated reporters. Notably, out of 17 different genetic screens performed using the same insertional mutagenesis approach in HAP1 cells, genes codifying NELF complex subunits resulted as significant hits only in the current LAD regulator screen (Fig. EV3A).The NELF complex is known to inhibit transcription elongation in vitro (Narita et al, 2003) and has been proposed to contribute to the pausing of RNA polymerase II (PolII) just downstream of promoters, although its role in vivo is still a matter of debate (Aoi and Shilatifard, 2023).

### DSIF also acts as repressor in LADs

NELF exerts its inhibitory function in association with DSIF (DRB Sensitivity-Inducing Factor) (Decker, 2021; Wu et al, 2003). Our data indicate that SUPT4H1 (also known as SPT4), one of the two subunits of DSIF, also acts as strong repressor of both LAD reporters (MI = 3.56 for LAD5; MI = 2.73 for LAD6). The other DSIF subunit, SUPT5H/SPT5, showed slightly positive but non-

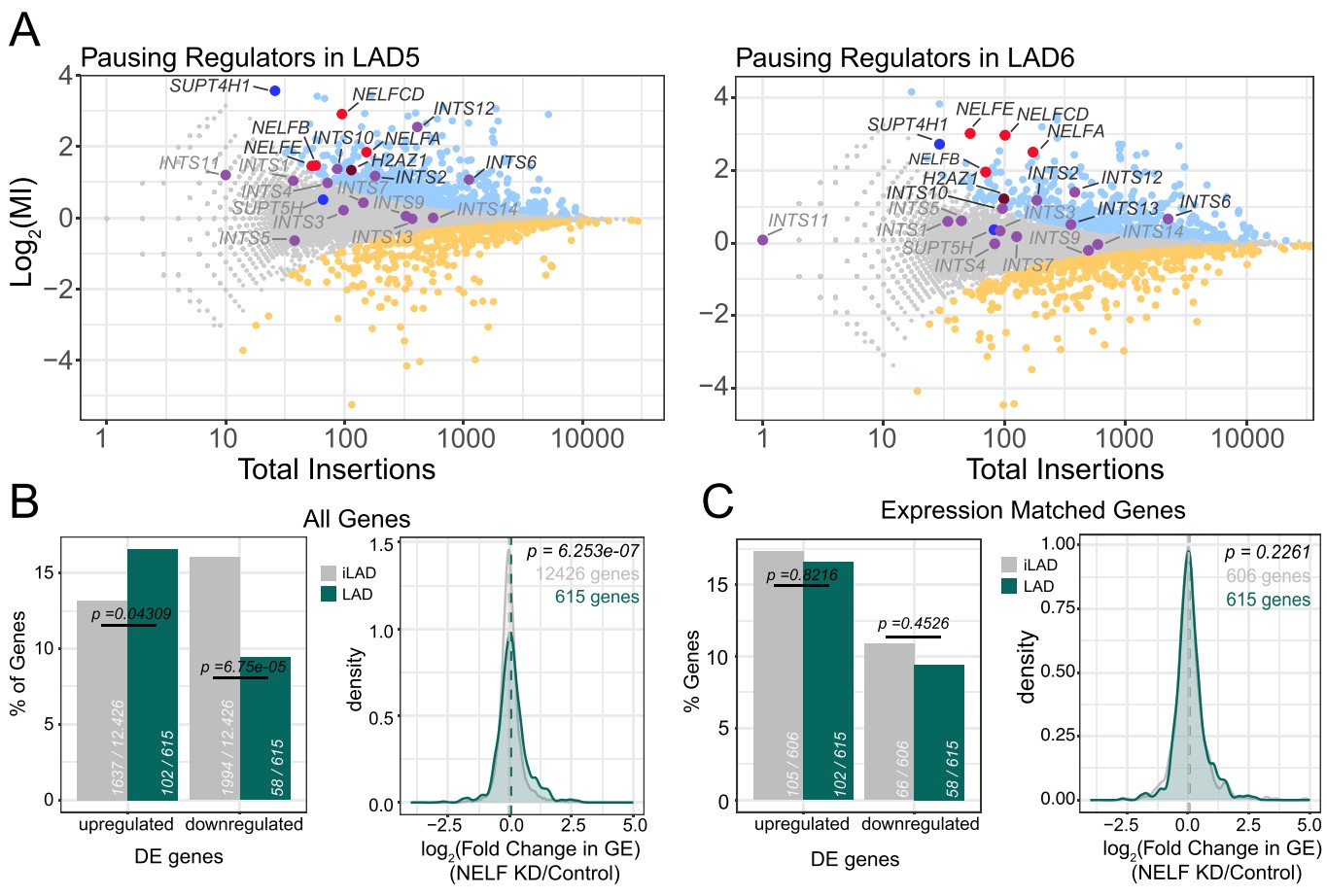

**Figure 3. Screens identify regulators of PolII elongation as repressors in LADs.**

(A) Mutational indices for pausing regulators, in LAD5 (left panel) and LAD6 (right panel). Blue, LAD activators; orange, LAD repressors. Gray, non-significant genes. Several complexes or proteins involved in regulation of pausing are highlighted with different colors. (B) Left: Percentage of up- and down-regulated genes in LADs and iLADs after NELF knockdown. Numbers of genes analyzed in each class are indicated. Completely inactive genes (TPM < 0.3 in control sample) were not included in this analysis. Statistical significance was calculated according to Fisher's Exact Test. Right: density plot of $\log_2$ (fold change) in gene expression (GE) following NELF knockdown for all LAD and iLAD genes. *P* value according to Wilcoxon's test. (C) Same as (B), but using a subset of iLAD genes that matched LAD genes for expression levels. Data are from 2 replicates for NELFB knockdown and 3 replicates for NELFE knockdown which were combined together.

significant MI scores (Fig. 3A), possibly indicating that this subunit is less critical for the repressive action of DSIF/NELF in LADs.

### *Repressor hits include additional regulators of PolII pausing*

These findings prompted us to search the putative LAD repressors for additional regulators of PolII pausing. One of these is Integrator, which is known to terminate paused transcription and to regulate initiation and pause release (Elrod et al, 2019; Gardini et al, 2014). Integrator was shown to physically interact with both DSIF and NELF (Yamamoto et al, 2014). In line with this finding, five of the 14 subunits of this complex (INTS2, INTS6, INTS10, INTS12, INTS13) were among the putative LAD repressors, four of which were shared between LAD5 and LAD6 (Fig. 3A). Finally, the histone variant H2AZ1, which contributes to NELF recruitment at promoters and limits the transition from pausing to elongation (Mylonas et al, 2021), acted as a repressor in both LADs (MI = 1.33 for LAD5, MI = 1.22 for LAD6). Thus, in our screen for LAD repressors we identified multiple factors that are known to interact and cause promoter-proximal PolII pausing (Fig. 3A).

### **NELF acts as a mild repressor of genes in LADs genome-wide**

#### *Depletion of NELFB and NELFE*

Next, we investigated to what degree NELF could contribute to the suppression of gene activity in LADs throughout the genome. We transiently knocked down either NELFB or NELFE by CRISPR interference (CRISPRi) (Gilbert et al, 2014) in HAP1 cells. In each case we observed a strong depletion of the respective protein 6 days after transfection (Fig. EV3B). Depletion of NELFE also induced loss of NELFB protein (Fig. EV3B), in agreement with previous results indicating that the stability of the NELF subunits is interdependent (Narita et al, 2007).

#### *NELF knockdown affects a few thousand genes genome-wide*

Subsequent mRNA-seq analysis identified a few thousand genes that are either up- or downregulated after NELF depletion. The quantitative changes in expression of the significantly de-regulated genes showed a high correlation between the NELFB and NELFE

knockdowns (Fig. EV3C). We therefore combined the mRNA-seq data of these two knockdowns, which provides additional statistical power. This identified 2028 upregulated and 2348 downregulated genes following NELF depletion. However, the extent of regulation was relatively marginal, because only 475 upregulated and 177 downregulated genes showed more than 1.5-fold change. This indicates that NELF can act both as a repressor and activator, but for most target genes the regulatory effect is modest, in agreement with earlier studies (Aoi et al, 2020; Gilchrist et al, 2008).

### Subset of LAD genes is repressed by NELF

We then focused on 615 LAD genes with at least some detectable RNA-seq signal (>0.3 transcripts per million) in control cells, reasoning that these genes carried a functional transcription machinery and thus had the potential of becoming upregulated upon NELF depletion. Of these genes, 16.6% showed significantly increased expression upon NELF depletion (Fig. 3B, left panel). Thus, a substantial proportion of genes in LADs is repressed by NELF. Interestingly, this proportion was lower (13.2%) in iLADs. In contrast, genes with reduced expression upon NELF depletion were less frequent in LADs and more frequent in iLADs (9.4% in LAD out of 615 genes and 16.0% in iLAD out of 12,426 genes) (Fig. 3B, left panel). Analysis of the quantitative expression changes also showed that upregulation of LAD genes is more prevalent than downregulation following NELF depletion (Fig. 3B, right panel).

### Preference of NELF for LAD genes may relate to their transcriptional status

While these data may suggest that NELF preferentially represses genes in LADs, they need to be interpreted with caution because genes in iLADs are generally more active than genes in LADs, which could be a confounding factor. In order to correct for this, we repeated this analysis with a subset of iLAD genes selected to match the distribution of expression levels seen for all LAD genes (in the presence of NELF) (Fig. EV3D). In this subset, the proportions of up- and down-regulated genes in iLADs after NELF depletion are more similar to that of LADs (Fig. 3C, left panel). Analysis of the quantitative expression changes in LADs and the matching iLAD set confirms this result (Fig. 3C, right panel). Furthermore, analysis of available NELFE ChIP-seq data from HeLa cells (Beckedorff et al, 2020) indicates that promoters in LADs are not more strongly or frequently occupied by NELF than expression-matched promoters in iLADs (Fig. EV3E). Based on these combined results, we speculate that the apparent preference of NELF to repress genes in LADs may be related to the fact that genes in iLADs are generally more active and thereby more resistant to repression by NELF.

### NELF role in LADs depends on local context and promoter identity

The broad range of gene expression responses to NELF depletion in LADs (Fig. 3B,C) raised the question whether the effect of NELF depends on the promoter, on the local chromatin context, or both. To test this, we employed our TRIP technology, in which a barcoded reporter gene driven by a single promoter is inserted into a large number of random positions throughout the genome, both in LADs and iLADs (Akhtar et al, 2013; Leemans et al, 2019). We selected three previously established K562 cell pools with hundreds of integrated reporters driven by promoters of the *BRINP1*, *ARGHEF*, and *MED30* genes (Leemans et al, 2019). Using RNA interference, we then depleted NELFE and NELFB in these cell pools (Fig. EV4A) and measured resulting changes in expression

level for each reporter, split by LAD and iLAD integrations. Overall, the effects of NELFE and NELFB depletion again correlated strongly (Fig. EV4B). For the *MED30* promoter, NELF depletion induced on average a slightly higher upregulation in LADs compared to iLADs (Fig. EV4C). However, the response of this promoter was highly variable across integration sites, both in LADs and iLADs. A similar broad variation was obtained for reporters driven by the *ARGHEF* and *BRINP1* promoters; for these promoters we did not detect a systematic difference between LAD and iLAD integrations, although we note that the *BRINP1* promoter is so lowly active in most LADs (Leemans et al, 2019) that changes in activity upon NELF depletion could not be determined reliably for most LAD integrations. We conclude that the local chromatin context has a profound effect on the responsiveness of promoters in LADs to NELF, and that promoters may intrinsically differ in their sensitivity to NELF.

## Dual role of the BAF complex in controlling transcription of LAD reporters

### BAF has opposite effects on two LAD reporters

One of the most significantly enriched GO categories for the screen hits in LAD6 was "ATP-dependent chromatin remodeler activity" (Appendix Fig. S2C). This category was also significantly ($P = 2 \times 10^{-5}$) enriched in LAD5. Among the most significant repressor hits in LAD6 we identified several subunits of the chromatin remodeler complex BAF, such as the catalytic subunit SMARCA4 (also known as BRG1), and other subunits from different parts of the complex, including SMARCB1, SMARCC1, SMARCE1, and ARID1A (Fig. 4A). This suggested that BAF acts as a repressor in this LAD. Surprisingly, the same five subunits behaved as strong *activators* in LAD5 (Figs. 4B and EV4D). Because the reporters in LAD5 and LAD6 are driven by the same promoter, this must be due to differences in the local chromatin context (Fig. EV1C),

### PBAF may also be involved

Some of the effects (either repressive or activating) may also be attributed to the PBAF complex, which shares many subunits with BAF, except that it contains ARID2, BRD7, and PHF10 instead of ARID1A, BRD9, and DPF1/2/3, respectively (Alfert et al, 2019; Hodges et al, 2016). These PBAF subunits also show some effects in either LAD5 or LAD6, albeit more moderately than some of the BAF subunits (Figs. 4B and EV4D).

### BAF mediates up- and down-regulation of transcription in LADs genome-wide

The opposite effects of multiple BAF subunits on the reporter activities in LAD5 and LAD6 suggested that BAF could modulate genes in LADs both as a repressor and an activator, depending on the local context. To test if this is the case genome-wide, we analyzed available RNA-seq data from HAP1 cell lines in which SMARCA4, SMARCC1, or ARID1A were knocked out (Schick et al, 2019). Strikingly, depletion of each of these three subunits of BAF led to a higher percentage of both up- and down-regulated genes in LADs as compared to iLAD regions (Fig. 4C).

### Regulatory preference for LADs is not due to expression levels

As explained above, this result could be confounded by the fact that genes in iLADs are generally more active than genes in LADs. Generally, measurements of mRNA abundance are less noisy for

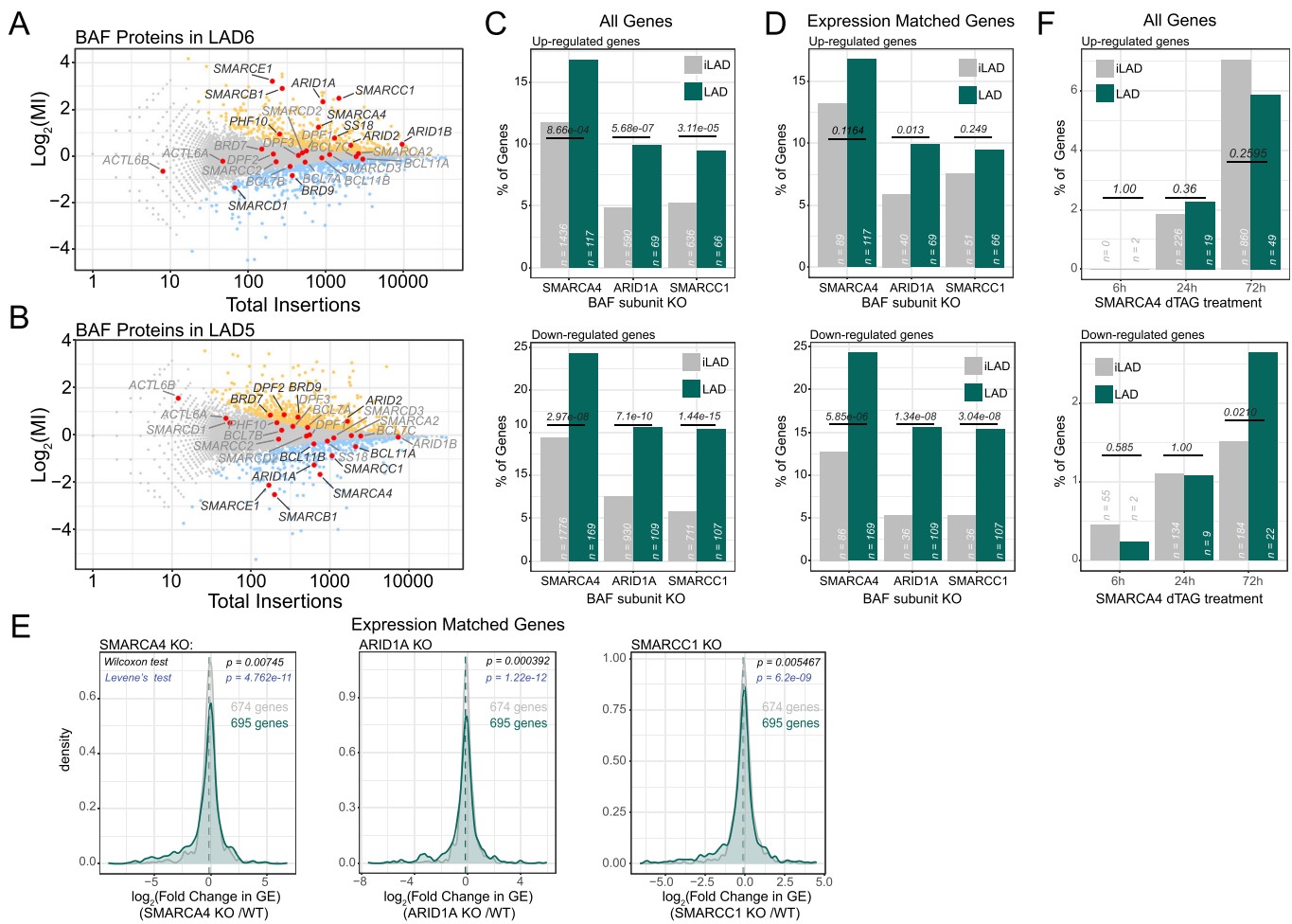

**Figure 4. BAF complex has an amplified role in LADs.**

(A, B) Screen results with BAF complex subunits highlighted in red. BAF mainly acts as a repressor in LAD6 (A) and mostly as an activator in LAD5 (B). Blue, LAD activators; orange, LAD repressors; gray, not significant; red, BAF subunits. (C) Percentage of up- and down-regulated genes in LADs and iLADs in SMARCA4, ARID1A, and SMARCC1 knockout cells. "n" indicates the number of genes analyzed in each class. Completely inactive genes (TPM < 0.3 in control sample) were not included in this analysis. (D) Same as (C), but using a subset of iLAD genes that matched LAD genes for expression levels. (E) Density plot of of log$_2$ (fold change) in gene expression (GE) following SMARCA4, ARID1A, and SMARCC1 knockout for LAD genes and expression-matched iLAD genes. Data are from (Schick et al, 2019). Results are from three biological replicates. Statistical significance was calculated with Wilcoxon test for comparison of median, and Levene's test for comparison of variance. (F) Same as (C), but for acute depletion of SMARCA4 (6, 24, and 72 h of depletion) (Schick et al, 2021). Results are from two biological replicates. For (C), (D), and (F) Statistical significance was calculated with Fisher's Exact test.

highly active genes than for lowly active genes; it is also possible that highly active genes are less dependent on BAF. However, comparison to a subset of iLAD genes that were selected to match the expression levels of LAD genes (Fig. EV4E) confirmed that LAD genes are more frequently up- and down-regulated than expression-matched iLAD genes after knockout of each of the three BAF subunits (Fig. 4D). Analysis of the quantitative expression changes after knockout also shows a broader distribution in LADs than in the matching iLAD set (Fig. 4E). We conclude that BAF preferentially controls gene activity in LADs and can act both as a repressor and as an activator, depending on the gene.

*Response to BAF loss appears to be slow*
To understand the kinetics of the regulation of LAD genes by BAF we analyzed available RNA-seq data following rapid dTAG-mediated SMARCA4 degradation (Schick et al, 2021). Only at

72 h after BAF depletion this led to a similar number of upregulated genes as the stable knockout, while the number of downregulated genes remained ~5-fold lower than in the knockout. We did not observe a preferential enrichment of de-regulated genes in LADs with the exception of a higher proportion of downregulated genes at 72 h (Fig. 4F). These data indicate that the differential response of LAD genes to BAF loss may be a slow process. This could indicate that the effect of BAF on LAD genes is indirect.

## The Mediator complex acts as activator and repressor in LADs

*Mediator subunits have opposite effects on the two LAD reporters*
Other striking hits in our screens were multiple subunits of the Mediator complex. This complex generally functions as a transcriptional activator, but some of its subunits can also act as

a repressor (Luyties and Taatjes, 2022). Out of 32 known Mediator subunits, we found that 15 act as repressors of the LAD6 reporter (Fig. 5A). This includes MED12, CCNC, and CDK8, which are subunits of the kinase module of Mediator, which has previously been linked to transcriptional repression (Pelish et al, 2015; Postlmayr et al, 2020). Of the subunits without significant effects, most lacked statistical power due to low overall numbers of insertions, which can be due to either the small size of the genes or to their essentiality. Remarkably, we found that the Mediator complex yielded almost perfect mirror-image results for LAD5: 14 out of the 15 Mediator subunits that act as repressors in LAD6, act as significant activators in LAD5 (Fig. 5B). Thus, similar to BAF, the effect of Mediator on reporter activity in LADs strongly depends on the local context.

Interestingly, analysis of available ChIP-seq data for a subunit of Mediator (MED26) (El Khattabi et al, 2019) suggests that promoters in LADs are rarely bound by Mediator, unlike promoters in iLADs (Fig. EV5A). In contast, enhancer-like elements are equally frequently bound by MED26 in LADs and iLADs (Fig. EV5A). The proximity of such enhancer-like elements to the reporter in LAD5 but not LAD6 (Fig. EV1C) may thus explain why the LAD5 reporter is selectively activated by Mediator. Possible explanations for the repressive activity in LAD6 are explored below.

### MED12 preferentially represses endogenous gene activity in LADs

To further investigate the impact of Mediator on transcription in LADs, we analyzed published RNA-seq data from HAP1 cells in which MED12 was deleted (Haarhuis et al, 2022). MED12 was among the top-5 of Mediator subunits with the strongest effect on reporter activity in the screens (Fig. 5A). Strikingly, HAP1 cells lacking MED12 showed upregulation of 27% of all genes in LADs with detectable RNA-seq signal. This percentage is almost twice the one observed for expression-matched genes in iLADs (15%; Figs. 5C and EV5B). In comparison, the percentage of downregulated genes in LADs was markedly lower (14%), and not significantly different from that in iLADs. Quantitative analysis of the changes in expression levels upon MED12 knockout show a broader distribution for LAD genes than for iLAD genes, with a tendency towards upregulation (Fig. 5D). These results indicate that MED12 is involved in the repression of a substantial proportion of genes in LADs, although at a lower frequency it can also activate genes in LADs. The LAD5 and LAD6 reporters thus may have captured both effects.

### Effect of Mediator depletion appears to be slow

To complement these results we analyzed recent gene expression data from KBM7 cells in which several subunits of the Mediator complex were depleted through the dTAG degron technology (Jaeger et al, 2020). As was reported in that study, only depletion of MED1 and MED14 caused marked changes in transcription, while the other tested subunits had virtually no effect (Appendix Table S1). By re-analysis of these data, across eight different Mediator subunit depletions, five showed significantly higher effects in LADs. However, there were no pronounced dissimilarities between LAD genes and iLAD genes, including two (MED1 and MED12) hits present in our screens (Fig. EV5C). While we cannot rule out that this discrepancy with the stable MED12 knockout results is due to cell-type specific differences (even though

HAP1 cells were derived from KBM7 cells (Carette et al, 2010)), these data suggest that the effects of loss of Mediator subunits on gene expression only develop slowly and may involve indirect mechanisms.

### Mediator may affect LAD gene activity by controlling H3K9me3 distribution

One possible explanation for the slow effects of Mediator could be a gradual reshaping of the epigenome. It was previously found that stable deletion of MED12 causes accumulation of H3K9me3 in large genomic regions, and depletion of this heterochromatin mark elsewhere in the genome (Fig. EV5D,E (Haarhuis et al, 2022)). Because H3K9me3 tends to be enriched in LADs (Fig. EV5F) (Briand and Collas, 2020; Hoskins et al, 2021; Manzo et al, 2022; van Steensel and Belmont, 2017), we considered that the changes in LAD gene expression may be due to this reshaping of the epigenome. We therefore determined the changes in H3K9me3 of up- and downregulated genes upon MED12 knockout. Strikingly, about half of downregulated genes in LADs showed a prominent gain of H3K9me3, while very few of expression-matched iLAD genes showed increased H3K9me3 levels (Fig. 5E, top panel). These data suggest that increased repression following the loss of MED12 can be explained, at least partially, by the concentration of heterochromatin in LADs. Upregulated genes generally showed a loss of H3K9me3, as may be expected. However, this H3K9me3 loss was not specific for LAD genes (Fig. 5E, bottom panel). Together, these results indicate that the reshaping of the H3K9me3 landscape (Haarhuis et al, 2022) may explain in part the regulatory effects of Mediator in LADs.

### MED12 controls patterns of putative enhancer activity in LADs

Mediator is important for enhancer function (Richter et al, 2022) and evidence has also been reported that partitioning of the genome at the nuclear periphery may constrain the function of enhancers (Robson et al, 2017; Smith et al, 2021). We therefore explored whether changes in enhancers may partially account for the altered gene expression in LADs after Mediator depletion. To address this, we scored genes that exhibited a gain or loss of the most proximal H3K4me1 peak (a marker of putative enhancers (Rada-Iglesias, 2018)) upon knockout of MED12 (Haarhuis et al, 2022). Remarkably, genes in LADs showed a higher frequency of such changes of nearby H3K4me1 peaks than genes in iLADs (Fig. 5F). Subsequent analysis of changes in gene expression in MED12 knockout cells revealed that gained H3K4me1 peaks generally correlated with transcriptional activation, while a loss of H3K4me1 peak paralleled transcriptional downregulation. These changes in gene expression were much more pronounced and consistent in LADs than in iLADs (Fig. 5G). This is true both for genes with lost and gained H3K4me1 peaks, although the latter consist only of a few tens of genes that are clustered mainly on a few chromosomal locations, such as ZNF and the PCDHB gene clusters. Together, these results indicate that MED12 not only plays a more prominent role in controlling H3K4me1 peaks near genes in LADs compared to iLADs; it also has more impact on the expression level of LAD genes with such dynamic H3K4me1 peaks. It is tempting to speculate that some of the effects of MED12 on gene activity in LADs are exerted via H3K4me1-marked enhancers, but other mechanisms cannot be ruled out.

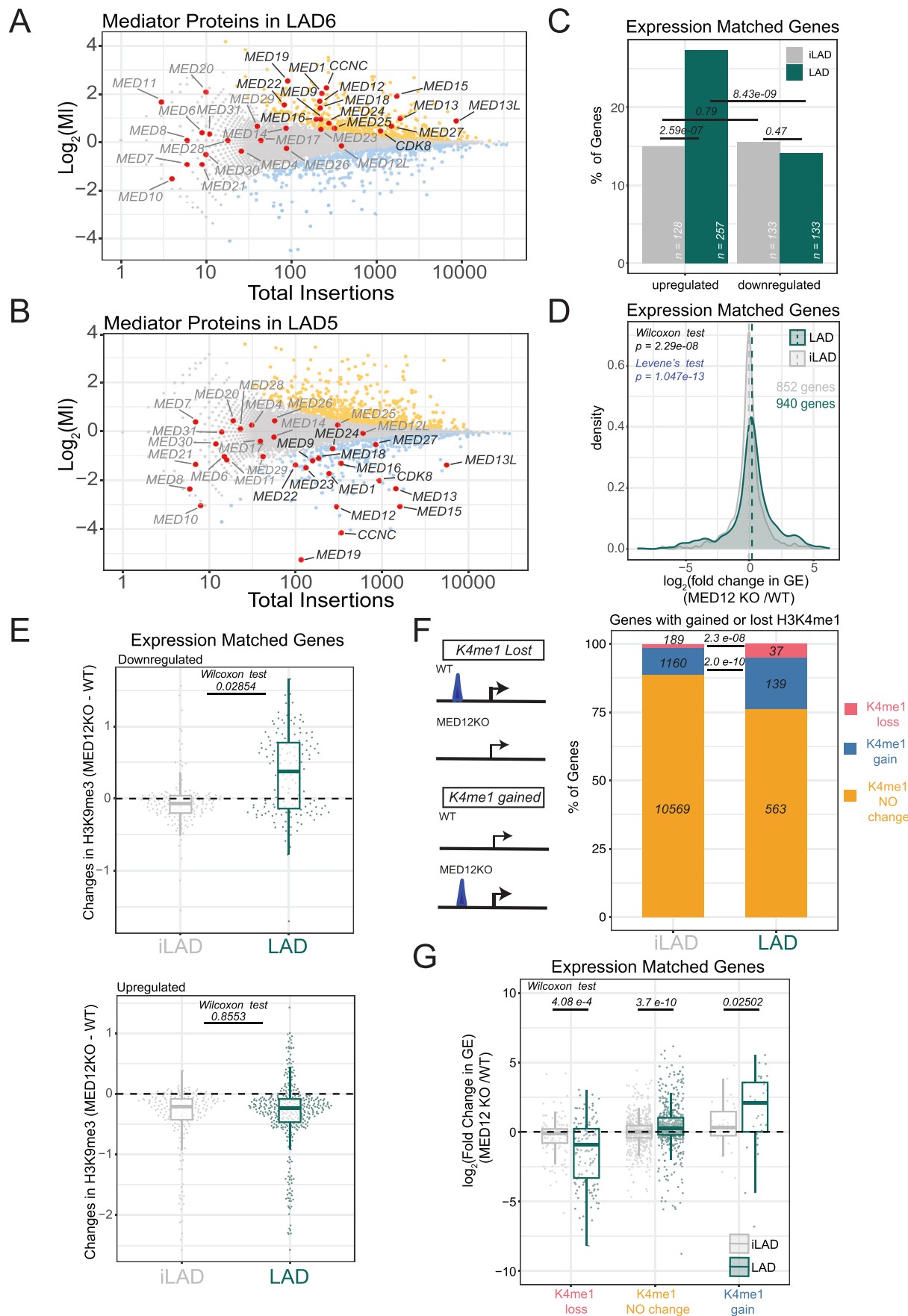

◄ **Figure 5. Mediator modulates heterochromatin and putative enhancers in LADs.**

(A, B) Screen results with Mediator subunits highlighted in red. Mediator mainly acts as a repressor in LAD6 (A) and mostly as an activator in LAD5 (B). Blue, LAD activators; orange, LAD repressors; gray, not significant; red, Mediator subunits. (C) Percentage of up- and down-regulated genes following MED12 knockout for LAD and expression-matched iLAD genes. "n" indicates the number of genes analyzed in each class. Completely inactive genes (TPM < 0.3 in control sample) were not included in this analysis. Statistical significance was calculated with Fisher's Exact test (D) $\log_2$(fold change) in gene expression (GE) following MED12 knockout for expression-matched LAD and iLAD genes. Statistical significance was calculated with Wilcoxon test for comparison of median, and Levene's test for comparison of variance. (E) Changes in H3K9me3 levels at genes for downregulated (top) and upregulated (bottom) LAD genes or expression-matched iLAD genes. (F) Left: cartoon depicting how gain and loss of H3K4me1 peaks following MED12 knockout were selected. Right: Proportion of iLAD and LAD genes with a gain or loss of proximal H3K4me1 peak following MED12 knockout. *P* values according to Fisher's Exact test. (G) $\log_2$(fold change) in gene expression (GE) following MED12 knockout for LAD and expression-matched iLAD genes, divided by gain or loss of proximal H3K4me1 peak following MED12 knockout. *P* values in (E, G) are according to Wilcoxon's test. For (G, E), the central line in the boxplots represents the median. The lower and upper hinges correspond to the first and third quartiles (the 25th and 75th percentiles). The upper and lower whiskers extends from the hinge to the largest or smallest values respectively no further than 1.5 * inter-quartile range. Outliers were removed only for visualization purposes. For (B, C–F, G), data are from (Haarhuis et al, 2022) and results are from 6 biological replicates.

## Discussion

### Chromatin proteins have more regulatory impact in LADs than NL proteins

One of the noteworthy results of our screens is that chromatin proteins appear to play much more prominent roles than NL/NE proteins in the repression of reporter activity in LADs. We cannot rule out that NL/NE proteins act more redundantly than chromatin proteins, and thus are missed in the screens. Some NL/NE proteins may also have been missed due to their essentiality. However, a more parsimonious interpretation is that the local chromatin composition is the main determinant of gene repression in LADs, whereas interactions with NL proteins are of lesser importance. We note that most LADs interact only stochastically with the NL, and can remain detached from the NL for an entire cell cycle (Kind et al, 2015; Kind et al, 2013). Single-cell analysis has indicated that genes in these LADs are only subtly less active when they contact the NL, compared to when they are in the nuclear interior (Rooijers et al, 2019). In this light, it makes sense that gene repression is not strongly dependent on NL proteins. It remains possible, however, that transient interactions with the NL add robustness to gene repression by chromatin proteins. An example of this is the LAD-embedded *ABCB1* gene (encoding a drug exporter), which can become derepressed in a small sub-population of cells when the NL protein LBR is deleted, leading to sporadic drug-resistant cell clones (Manjon et al, 2023). Our screen was not designed to pick up such rare, stochastic de-repression events.

### A prominent role for elongation regulators

Our screens identified multiple proteins and complexes that are known to inhibit transcription elongation or promote pausing of PolII close to the promoter. This included NELF and DSIF, which directly interact with PolII and the nascent transcript; the histone variant H2AZ1, which has been shown to hinder PolII progression at the pausing site; and the Integrator complex, which promotes early termination (Beckedorff et al, 2020; Gardini et al, 2014). This finding offers a mechanistic explanation for earlier observations that LADs are unfavorable to transcription elongation and show increased promoter-proximal pausing of PolII (Leemans et al, 2019; Wu and Yao, 2017). Our genome-wide analysis of gene expression after NELF depletion indicates that genes in LADs tend to be somewhat more frequently repressed by NELF than genes in iLADs.

However, available proteomic data does not indicate that NELF is enriched near the NL or at LAD chromatin (Cheng et al, 2023; Wong et al, 2021). We propose that the repressive activity of NELF in LADs is not related to proximity to the NL, but rather reflects a preference of NELF to act on relatively lowly active genes, which are more prevalent in LADs. Nevertheless, it is striking that our screens identified a broad diversity of proteins that have been implicated as elongation regulators. It is possible that these factors act synergistically to repress transcription in LADs. Thus, the collective contribution of pausing factors, nucleosome barriers, and termination factors could enhance pausing, block elongation, and favor early termination preferentially in LADs.

### Context-dependent bi-functionality of BAF in LADs

We found that BAF acts as a repressor in LAD6, and as an activator in LAD5. Because the same reporter was used in both LADs, this implies that the role of BAF is dependent on the local context. Our genome-wide analysis of LADs underscores this bidirectional role of BAF in LADs, and indicates that this occurs more prominently in LADs than in iLADs. BAF has been shown to regulate accessibility at enhancers and superenhancers (McDonald et al, 2023; Schick et al, 2021) and this could explain the activator role described for LAD5. BAF can also regulate the binding of repressors: for example long-term inhibition of BRG1 causes reduction in binding of repressor REST (Iurlaro et al, 2021). In line with these observation we found that both REST and its co-repressor RCOR1 were weak repressors of both LAD5 and LAD6 reporters, suggesting these repressors might have a preferential role in LADs. It remains to be elucidated which local feature in LADs determines whether BAF activates or represses gene activity and if this happens at enhancer or promoters levels. However, differences in local chromatin composition that we identified observed for the two LADs are likely to occur across the more than 1000 LADs in mammalian genomes.

### Bi-functionality of Mediator in LADs

We observed a similar bi-functionality of Mediator. While this complex is mostly known as an activator, a repressive role is not unprecedented; for example, it has been linked to Xist-mediated repression (Postlmayr et al, 2020) and its kinase module subunits were reported to restrain activation of genes by a super-enhancer (Pelish et al, 2015). Our analyses suggest that the repressive and the

activating effects may be explained in part by the re-distribution of H3K9me3-marked heterochromatin within LADs that is triggered by the depletion of Mediator (Haarhuis et al, 2022). Modulation of the activity of nearby enhancers may also play a role. Different kinetics of loading and module association/dissociation for the Mediator complex might also be differently regulated between LAD and iLADs, and this could contribute to LAD repression. We have found that MED26 is not efficiently recruited at LAD promoters. Due to limited availabililty of Mediator ChIP-seq data (El Khattabi et al) we can only speculate on a possible mechanism. The kinase module (MKM) needs to dissociate from the rest of the complex to allow direct interaction with RNA polymerase (Richter et al, 2022). Since MKM is mutually exclusive with MED26, the lack of this subunit from LAD promoters might suggests that the mediator kinase module (MKM) does not dissociate at LAD enhancers and, therefore, impedes the formation of a transcription-competent mediator at LAD promoters. This model could also explain why the subunits of the MKM are among the most significant hits in our screen. Understanding if this is a possible mechanism will require further studies.

## All the above factors might be linked in regulating pausing of PolII at LADs

Possibly, the roles of the elongation regulators, BAF and Mediator in LADs are intertwined. A recent study found PolII pausing to stabilize BAF on chromatin and to enhance its nucleosome eviction activity (Brahma and Henikoff, 2024). Similarly, Mediator (and especially its kinase module) has been linked to regulation of PolII pausing and elongation (Luyties and Taatjes, 2022; Richter et al, 2022). Further studies will be required to understand more deeply how the interplay between these factors orchestrates transcription in LADs.

## Limitations of the study

Our screens were based on stable knockout of individual proteins over several days. Thus, we cannot rule out indirect effects and cellular adaptation as possible explanations for some of the hits. Indeed, short-term effects upon degron-mediated depletion of BAF and Mediator subunits were generally undeniably weaker than those observed with stable depletions. This could mean that BAF and Mediator do not act directly on genes in LADs, but rather regulate another process that in turn affects genes in LADs. An example is the reshaping of H3K9me3 upon MED12 loss, which may in turn affect the expression of genes in LADs. Further studies are needed to elucidate the precise mechanisms by which NELF, BAF, Mediator and other hits from the screen control gene activity in LADs. While many hits were shared between the LAD5 and LAD6 reporters, the two screens also showed many differences and —as examplified by Mediator and BAF subunits—even opposite effects. We cannot rule out that this is due to the fact that the LAD6 reporter was integrated as an array of multiple copies. This could lead to complex cross-talk within the array, or affect the propensity of the locus to adopt a heterochromatic state. However, the reporter integration sites of LAD5 and LAD6 also differ markedly in their local chromatin makeup. Previous analyses of identical reporters integrated in hundreds of LADs has underscored that the local chromatin environment within LADs strongly affects gene activity (Leemans et al, 2019), and our genome-wide analyses also point to

strongly varying responses of genes in LADs to perturbations of NELF, BAF and Mediator. Regulation of genes in LADs thus appears to be highly dependent on the local context, and thus may be controlled by a large diversity of chromatin regulators, rather than by universal "master" regulators. Our screen results provide a first view of this diversity.

# Methods

**Reagents and tools table**

| Reagent/Resource | Reference or Source | Identifier or Catalog Number |
|---|---|---|
| **Experimental Models** | | |
| HAP1 | | Prof Thijn Brummelkamp |
| HAP1 KRAB-dCAS9 | This study | N/A |
| HeLa | Netherlands Cancer Institute | Prof Rene' Medema |
| HCT116-Top2A-mAID | | Prof Vassilis Roukos |
| **Recombinant DNA** | | |
| pX330 U6-Chimeric_BB-CBh-hSpCas9 | Addgene | 42230 |
| pBABE puro | Addgene | 1764 |
| UCOE-EF1a-dCas9-BFP-KRAB-P2A-BlastR | Addgene | 118154 |
| pHDM-Hgpm2 | Addgene | 164441 |
| pHDM-Tat1b | Addgene | 164442 |
| pHDM-G | Addgene | 164440 |
| pRC-CMV-Rev1B | Addgene | 164443 |
| Lentiguide Puro | Addgene | 52963 |
| pMDL | Addgene | 12251 |
| pVSV-G | Addgene | 138479 |
| pRSV-Rev | Addgene | 12253 |
| **Antibodies** | | |
| NELFE | Merck-Millipore | ABE48 |
| TOP1 | Abcam | ab109374 |
| LMNB2 | Abcam | ab8983 |
| **Oligonucleotides and other sequence-based reagents** | | |
| Primers | This study | Dataset EV1 |
| NELFB On Target Plus smartpools | Dharmacon/Horizon | L-015839-01-0005 |
| NELFE On Target Plus smartpools | Dharmacon/Horizon | L-011761-01-0005 |
| Silencer™ Select Negative Control No. 2 siRNA | ThermoFisher | 4390846 |
| **Chemicals, Enzymes and other reagents** | | |
| Phusion® HF DNA Polymerase | New England BioLabs | M0530L |
| MyTaq Red Mix 2x | Bioline | BIO-25044 |
| CleanPCR | CleanNA | CPCR-0500 |
| X-fect | Takara Bio, 631318 | 631318 |

| Reagent/Resource | Reference or Source | Identifier or Catalog Number |
|---|---|---|
| FuGENE HD Transfection Reagent | Promega | E2311 |
| Lipofectamine RNAiMAX Transfection Reagent | ThermoFisher | 13778150 |
| Opti-MEM | GIBCO | 31985047 |
| IMDM | GIBCO | 21980032 |
| McCoy's 5A (Modified) Medium | GIBCO | 16600082 |
| Tn5 enzyme | Schep et al, 2021 | N/A |
| PEG 8000x | Sigma | P1458 |
| TAPS-NAOH | Sigma | T5130 |
| dimethylformamide | Sigma | D4551 |
| Fetal Bovine Serum | Sigma | F7524 |
| T4 Ligase | Roche | 10799009001 |
| ISOLATE II Genomic DNA Kit | Bioline | BIO-52067 |
| spermidine | Sigma | S0266 |
| digitonin | Merck-Millipore | 300410 |
| cOmplete Protease Inhibitor Cocktail | Roche | 11873580001 |
| SAM | New England BioLabs | B9003S |
| Dam | New England BioLabs | M0222L |
| spermidine | Sigma | S0266 |
| DpnI | NEB | R0176L |
| **Software** | | |
| Bowtie2 v2.3.4 | | http://bowtie-bio.sourceforge.net/bowtie2/index.shtml |
| Samtools v1.5 | | RRID:SCR_002105; https://www.htslib.org/ |
| Cutadapt v1.9.1 | | RRID:SCR_011841; https://cutadapt.readthedocs.io/en/stable/ |
| FastQC v0.11.6 | Babraham bioinformatics | https://github.com/s-andrews/FastQC/r |
| STAR 2.5.4a | Dobin et al, 2013 | NA |
| DEseq2 | Love et al, 2014 | NA |
| RStudio Server Version 1.3.1073 | RStudio Team | https://rstudio.com/ |
| R version 3.6.3 (2020-02-29) | R Core Team | https://www.r-project.org/ |
| ggplot2 | | https://ggplot2.tidyverse.org |
| **Other** | | |
| PCR Isolate II PCR and Gel Kit | Bioline | BIO-52060 |
| Tetro Reverse Transcriptase | Bioline | BIO-65050 |
| PureLink HiPure Plasmid Midiprep Kit | Invitrogen | K210004 |

| Reagent/Resource | Reference or Source | Identifier or Catalog Number |
|---|---|---|
| RNAeasy mini kit | QIAGEN | 74104 |
| RNase-Free DNase Set | QIAGEN | 79254 |
| Amicon Ultra 15 Filters | Merck-Millipore | UFC910024 |

## Cell lines and culturing conditions

All cell lines were grown in a humidified incubator at 5% $CO_2$ at 37 °C. The cell lines were routinely tested for mycoplasma contamination. HAP1 cells were grown in IMDM supplemented with 10% FBS, 1% penicillin/streptomycin, and 4 mM glutamine. HCT116 cells were grown in McCoys's modified medium, supplemented with 10% FBS, 1% penicillin/streptomycin, and 2 mM glutamine. HeLa cells were grown in DMEM medium, supplemented with 10% FBS, 1% penicillin/streptomycin, and 2 mM glutamine. K562 TRIP cells were grown in IMDM medium supplemented with 10% FBS, 1% penicillin/streptomycin, and 2 mM glutamine.

## Generation of LAD reporter cell lines

We transfected HAP1 cells with 2 μg of pX330 U6-Chimeric_BB-CBh-hSpCas9 (Addgene plasmid #42230; (Cong et al, 2013)) containing LAD-specific or AVVS1-specific guide RNAs, 1 μg of a repair donor cassette, and 100 ng of pBABE puro plasmid (Addgene plasmid #1764; (Morgenstern and Land, 1990)) using X-fect (Takara Bio, #631318). The PCR cassettes contained the BRINP1 promoter driving either EGFP or mCherry genes, and 50 nucleotides of homology arms specific for the targeted location (LAD5, chr6:48,829,698; LAD6, chr7:118,951,271; AAVS1, chr19: 55,115,771). We generated the repair donor cassette by PCR using long primers with a phosphorothioate bond between the first and the third base (5′-3′) to minimize the degradation of the PCR product by cellular endonucleases (LongPrimerGFPLAD5_Fw, LongPrimerGF-PLAD6_Fw, LongPrimer_mCherryDBC1_AAVS1HA_Fw, see Dataset EV1). Thirty-six hours after transfection, we selected cells for puromycin resistance for 48 h. We FACS-sorted cell clones in 96 wells using side and forward scattering to select haploid HAP1 cells. We selected clones with integration by genotyping by PCR using primers at the 5′ junction between the BRINP1 promoter and the selected LAD/AAVS1 integration site. We further genotyped selected clones with different primer combinations to detect WT allele, 5′ and 3′ integration sites, and possible multiple tandem, divergent, or convergent integrations (Appendix Fig. S1A,B). Amplification over the full integration was problematic in both clonal cell lines, probably due to the high GC content in the BRINP1 promoter. Using primers that anneal only on the reporters revealed tandem unidirectional integrations for the LAD 6-GFP reporter and for the mCherry reporter in AAVS1 locus in both LAD5 and LAD6 cell lines.

## Mapping of reporter integrations

We performed TagMap as previously described (Stern, 2017) with minor modifications (Schep et al, 2021). Briefly, we tagmented 100 ng of genomic DNA from clones with transposome (Tn5 loaded with oligo adapters), followed by a linear amplification using a primer that annealed on the BRINP1 promoters or the

end of either *EGFP* or *mCherry* genes. 5 µl of linear amplified material was further amplified with two rounds of nested PCRs to introduce barcode and Illumina Adapters (PCR1 and PCR2). PCR1 was performed using 2 µl of enrichment PCR and *Illumina primer (N5xx)* and *BadAdap_DBC1_Rv_2* or *BadAdap_mCherryend_Fw* or *BadAdap_GFPend_Fw*. 2 µl of PCR1 were used as template for PCR2 using *Tn5_PCR2_R* and *Illumina_Nextera_N7xx* (N7xx) (see Dataset EV1). We used MyTaq Red Mix (Bioline, # BIO-25043) for all PCR reactions. We sequenced barcoded-indexed material on the MySeq Illumina Platform, with Paired-end 150 bp long reads. Adapters sequences were trimmed off from reads with a modified version of Cutadapt and trimmed reads were mapped to hg38 genome as well as the GFP and mCherry constructs.

reads from each population (see gating) are aligned to the hg38 human genome with zero or one mismatch using Bowtie. The reads were then assigned to protein-coding genes, considering the most extended open reading frame transcript, excluding overlapping regions that cannot be attributed to a single gene. Unique disruptive integrations (sense insertions in introns and exons) were counted between the transcription initiation site and stop codon. We calculated the mutational index (MI) by comparing normalized sense insertions of the different populations using a two-sided Fisher's exact test as previously described (Brockmann et al, 2017).

## Calculation of mutational indices

The mutational index (MI) was calculated as follows:

$$\text{Mutational Index(MI)} = \frac{\frac{\text{Number of sense insertions in genes in High population}}{\text{Total Number of sense insertions in High population} - \text{Number of sense insertions in genes in High population}}}{\frac{\text{Number of genic in genes insertions in Low population}}{\text{Total Number of sense insertions in Low population} - \text{Number of sense insertions in genese in Low population}}}$$

Both TagMap and genotyping PCR did not show amplification between mCherry and *AVVS1* for the LAD5-GFP cell line despite the AVVS1 locus being disrupted (Appendix Fig. S1A–C).

## FACS-based haploid genetic screen

We performed the screens as described previously (Blomen et al, 2015; Brockmann et al, 2017). Briefly, HAP1 cells were mutagenized with gene-trap retrovirus produced using HEK293T cells transfected with packaging plasmids (Gag-pol, VSVg, and pAdv) and a gene-trap vector containing BFP as a marker to assess the efficiency of the mutagenesis (Mazouzi et al, 2023; Blomen et al, 2015). 48 and 72 h after transfection, the supernatant was harvested, filtered, and concentrated using Amicon filters and stored at 4 °C. We combined both harvests, supplemented with protamine sulfate (8 µg/ml), and used them to infect $40 \times 10^6$ LAD5-GFP or LAD6-GFP HAP1 cells. The cells were then expanded to $5 \times 10^9$, harvested with trypsin, resuspended, pelleted, and washed once with PBS (phosphate buffered saline). Next, the cells were fixed with 4% formaldehyde (alcohol-free) for 10 min at RT, washed once with PBS containing 10% FCS (FACS buffer), and permeabilized at RT (room temperature) with 0.25% Triton X-100 in FACS buffer for 30 min. After PBS wash, the cells were stained for 1 h at RT with DAPI to visualize the G1 population. Finally, we filtered cellular suspension through a 40 µm strainer (BD Falcon). Next, the G1-phase cells were sorted using a BD FACSAria™ Fusion Cell Sorter into two bins based on their mCherry levels: low and high. Each bin was then further sorted based on the 4% highest and lowest GFP signal, resulting in four populations for each cell line (~11 million cells per population, see gating strategy in Appendix Fig. S2A). To obtain these cells, we processed 10 billion cells through sorting for both cell lines (LAD5 and LAD6), which required about 80 x T175 flasks. This process took 5 days of sorting for roughly 10 h per day, totaling ~50 h for both the LAD5 and LAD6 screens. Genomic DNA was isolated using a DNA mini kit (Qiagen). Subsequent sequencing of DNA libraries (about 800 million reads per screen), data analysis, and measuring of the insertions were performed as described previously with some modifications (Blomen et al, 2015; Brockmann et al, 2017; Mazouzi et al, 2023). In short, deep sequencing

We used a Two-sided Fisher's exact test adjusted for multiple testing using the Benjamini–Hochberg false discovery rate correction method for statistical significance. For calculation of the combined MI, the fastaq file from LOW1 was combined with LOW2 and HIGH1 with HIGH2.

## NELF CRISPRi and evaluation of NELF knockdown

To generate HAP1 cells stably expressing dCas9-BFP-KRAB, HAP1 cells were transduced with a lentiviral vector expressing dCas9-BFP-KRAB from an EF1-a promoter with an upstream ubiquitous chromatin opening element (UCOE-EF1a-dCas9-BFP-KRAB-P2A-Blast). We produced lentiviral particles by cotransfecting HEK293T cells in a 10 cm dish with lentiviral packaging plasmids pHDM-Hgpm2, pHDM-Tat1b, pHDM-G, and pRC-CMV-Rev1B, using FuGENE HD Transfection Reagent (Promega, E2311) following manufacturer's instructions. Supernatants were collected 48 h after transfection and filtered with a 0.45 µm filter. HAP1 cells were directly infected with 10–900 µl of filtered virus, cultured for 10 days, and pooled for sorting. BFP-positive cells were sorted on a BD FACS Aria Fusion and single cell plated to obtain a monoclonal cell line. This cell line was infected with lentiviral vectors (Lentiguide Puro: Addgene #52963) containing guide RNA for NELFE or NELFB. We produced lentiviral particles by transfecting HEK293T with calcium phosphate transfection and using pREV, pMDL, and pVSV as packaging vectors. Supernatants from a 10 cm dish were collected 48 h after transfection, filtered, and concentrated to 250 µl using Amicon filters, aliquoted, and stored at −80 °C. HAP1 cells were infected with 30 µl of concentrated virus and then selected for puromycin resistance 48 h after transduction. We performed mRNA extraction 144 h following lentiguide transduction for RNAseq and RT-qPCR. For protein extraction, nuclear proteins were extracted with a high salt buffer on isolated nuclei at different timepoints following transduction. Western blot was performed according to standard procedures using the following antibodies: NELFE (1:500, Merck-Millipore, ABE48), Top1 (1:5000, Abcam, ab109374).

## RNA-seq

RNA was extracted using an RNAeasy mini kit from Qiagen (#74104). One million cells were harvested, washed once in cold PBS, resuspended in 600 μL of RLT lysis buffer, and stored at −80 °C. Libraries were prepared using the TruSeq® RNA LT kit and TruSeq RNA Single Indexes (Illumina). We sequenced libraries with single-end 65-bp reads on a HiSeq 2500 platform. We sequenced ~30 million reads for every condition. Three independent biological replicates were generated for NELFE depletion and two for NELFB knockdown. RNA-seq reads were subjected to quality control using FastQC v0.11.6. Reads were aligned to the human reference genome (GRCh38, GRCh38_no_alt_analysis_-set_GCA_000001405.15; https://www.encodeproject.org/files/GRCh38_no_alt_analysis_set_GCA_000001405.15/) using STAR 2.5.4a (Dobin et al, 2013) with parameters --clip5pNbases 0 --outWigStrand Unstranded. Gene-level count tables were generated while mapping using Gencode v24 primary assembly annotations.

## DamID and pA-DamID

LMNB1 DamID data for HAP1 were previously generated (van Schaik et al, 2022) and are available from the 4DNucleome portal (https://data.4dnucleome.org/publications/5a5117c9-face-4648-bbb7 d54e468af2ba/#overview).

LMNB2 pA-DamID maps for the HAP1 dCAS9-KRAB clonal cell line, HCT116 and Hela were generated as previously described (van Schaik et al, 2022). Briefly, one million of cells were collected by centrifugation ($500 \times g$, 3 min) and washed sequentially in ice-cold PBS and digitonin wash buffer (D-Wash) (20 mM HEPES-KOH pH 7.5, 150 mM NaCl, 0.5 mM spermidine, 0.02% digitonin, Complete Protease Inhibitor Cocktail). Cells were rotated for 2 h at 4 °C in 200 μL D-Wash with 1:200 LMNB2 (Abcam, ab8983) followed by a wash step with D-Wash. Following this, cells were incubated with a solution of D-Wash buffer and Rabbit-anti-mouse IgG (1:200, Abcam, ab6709), followed by a wash step. This incubation was repeated with a 1:200 pA-Dam solution (equivalent to nearly 60 Dam units, determined by calibration against Dam enzyme from NEB, #M0222L), followed by two wash steps. Dam activity was induced by incubation for 30 min at 37 °C in 100 μL D-Wash supplemented with the methyl donor SAM (80 μM) while gently shaking (500 rpm). Genomic DNA was isolated and processed similarly to DamID, except that the DpnII digestion was omitted, and 65-bp reads were sequenced. For every condition, another 1 million cells were processed in only D-Wash and, during Dam activation, incubated with 4 units of Dam enzyme (NEB, M0222L). We use Dam-only control samples to normalize for DNA accessibility and amplification biases as described (Vogel et al, 2007).

## Analysis of RNA-seq for LAD and iLAD genes

RNA seq data for MED12 KO, BAF KO and BAF and Mediator degron experiments were downloaded from GEO: GSE125672 (Haarhuis et al, 2022); GSE108390 (Schick et al, 2019); GSE108390 (Schick et al, 2021); GSE139468 (Jaeger et al, 2020). For NELF depletion, RNA-seq was generated as mentioned above. DEseq2 (Love et al, 2014) was used to identify differentially expressed genes (adjusted *P* values < 0.05). In order to call LAD and iLAD genes, LMNB1-DamID or LMNB2-pA-DamID

generated in HAP1 cell lines from three independent biological replicates were used to calculate a genic NL association score. This score is the mean of LMNB1/Dam or LMNB2/Dam $\log_2$ ratio calculated over the gene length $+/-10$ kb. *Genes* with a DamID score >0 were defined as LAD genes. All the analyses were focused on genes with a minimal detectable RNA-seq signal by filtering out non-expressed genes (transcript per million, TPM < 0.3). Matching for gene expression was performed by calculating TPM for each gene and selecting a set of iLAD genes that matched LAD genes for TPM levels. The match function sorts genes according to expression levels and goes down this ordered list in blocks of 20. It then looks for the number of LAD genes in that set of 20 genes and randomly selects an equal number of iLAD genes.

## NELF depletion in K562 TRIP cell pools

Cell pools carrying integrated barcoded reporters were previously described (Leemans et al, 2019). For each pool, $5.4 \times 10^5$ cells were transfected in a 10 cm-dish with siRNA at final concentration of 10 nM and 1:1000 of Lipofectamin RNAimax reagent (Thermo-Fisher, 13778075). 48 h after transfection cells were expanded in three 10 cm-dish and transfected again in the same conditions. All experiments were performed at 72 h following the second round of transfection. Knockdown efficiency was checked by western blot analysis. As negative control siRNA we used Silencer™ Select Negative Control No. 2 siRNA (ThermoFisher, 4390846). For NELF depletion we used On Target Plus smartpools from Horizon Discovery (Dharmacon, NELFB,L-015839-01-0005; NELFE, L-011761-01-0005). Genomic DNA (gDNA) and RNA were extracted 72 h following the second round of transfection and used for library preparation as previously described (Leemans et al, 2019).

Computational analysis of *NELFE* and *NELFB* knockdown on TRIP libraries was performed using the same pipeline and LAD annotations as previously (Leemans et al, 2019). After analyzing the data quality of each of the samples, NELFB knockdown replicate 3 of the $P_{ARHGEF9}$ TRIP library and NELFB knockdown replicate 2 of the $P_{MED30}$ TRIP library were discarded due to low cDNA read counts.

For all other samples, barcode expression values normalized by gDNA was used for further analysis, using only barcodes with a gDNA count >100 in all samples. With these barcodes, first the correlation between replicates was compared to the correlation between NELFE and NELFB knockdown samples of the same replicate ($\rho = 0.647$ and $\rho = 0.690$ on average for replicates and between NELFE and NELFB conditions, respectively). $\log_2$(fold Change) in expression was calculated between NELF depletions and control for all barcodes. Wilcoxon test was used to determine significance of the difference il $\log_2$(fold Change) between LAD and iLAD barcode locations.

## Analysis of ChIP-seq data for LAD and iLAD enhancers and promoters

ChIP-seq data for NELFE were downloaded from GEO: GSE125534 (Beckedorff et al, 2020). ChIP-seq from MED26 were downloaded from GEO: GSE121355 (El Khattabi et al, 2019). Promoters were defined as previously described (Leemans et al, 2019) and a weighted mean ChIP score was calculated for a window covering 500 bp upstream and downstream of TSS. For HeLa, active promoters were selected by filtering for TPM > 0.3. For

HCT116 active promoters were filtered by TT-seq score >0.3. For enhancers, we used a list of annotated enhancers in HCT116 (Lidschreiber et al, 2021) to calculate a weighted mean score for MED26 binding.

## Data availability

RNAseq and pA-DAMID data generated in this work are available on GEO (accession code GSE261955). TagMap data (bam files) and TRIP data are available on SRA (accession code PRJNA1089502). Gene Trap screen raw data are available on SRA (accession code PRJNA1087589). Processed Gene Trap data are available at phenosaurus.nki.nl. Labnotes and R scripts regarding this study can be found at https://osf.io/z9cu8.

The source data of this paper are collected in the following database record: biostudies:S-SCDT-10_1038-S44318-024-00214-1.

## Peer review information

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

## Acknowledgements

We thank L. Gerace for sharing NE proteomics data; Elzo de Wit and Moreno Martinovic for sharing MED12 knockout data and ChIP-seq chromatin data in HAP1 before publication; the NKI Genomics, Research High-Performance Computing and the Robotics & Screening core facilities for technical support; and members of our laboratories for stimulating discussions; Federico Comoglio and Vinicius Franceschini-Santos for sharing R scripts, and Ludo Pagie for performing preliminary analysis. We thank Lise Dauban for critical reading of the manuscript, and members of our laboratory for helpful discussions. This work was supported by the European Union (ERC, GoCADiSC, 694466 to BvS; AIRC-MSCA iCARE2 fellowship, 800924, MSCA-IF 838555, and Next generation EU-MUR MSCA Young Researcher to SGM; ERC, CohesinLooping 772471, to BR); BR is also supported NWO, VI.C.202.098; AM is supported by an EMBO Long Term Fellowship (ALTF 158-2018). Views and opinions expressed are those of the authors only and do not necessarily reflect those of the European Union or the European Research Council. Neither the European Union nor the granting authority can be held responsible for them. Research at the Netherlands Cancer Institute is supported by an institutional grant of the Dutch Cancer Society and of the Dutch Ministry of Health, Welfare and Sport. The Oncode Institute is partially funded by the Dutch Cancer Society.

## Author contributions

**Stefano G Manzo**: Conceptualization; Resources; Data curation; Formal analysis; Supervision; Funding acquisition; Validation; Investigation; Visualization; Methodology; Writing—original draft; Project administration; Writing—review and editing. **Abdelghani Mazouzi**: Conceptualization; Data curation; Investigation; Visualization; Methodology. **Christ Leemans**: Software; Formal analysis; Methodology. **Tom van Schaik**: Software; Formal analysis. **Nadia Neyazi**: Validation; Investigation; Methodology. **Marjon S van Ruiten**: Investigation; Methodology. **Benjamin D Rowland**: Funding acquisition. **Thijn R Brummelkamp**: Conceptualization; Supervision; Funding acquisition; Methodology. **Bas van Steensel**: Conceptualization; Supervision; Funding acquisition; Writing—original draft; Project administration; Writing—review and editing.

Source data underlying figure panels in this paper may have individual authorship assigned. Where available, figure panel/source data authorship is listed in the following database record: biostudies:S-SCDT-10_1038-S44318-024-00214-1.

## Disclosure and competing interests statement

The authors declare no competing interests.

# Expanded View Figures

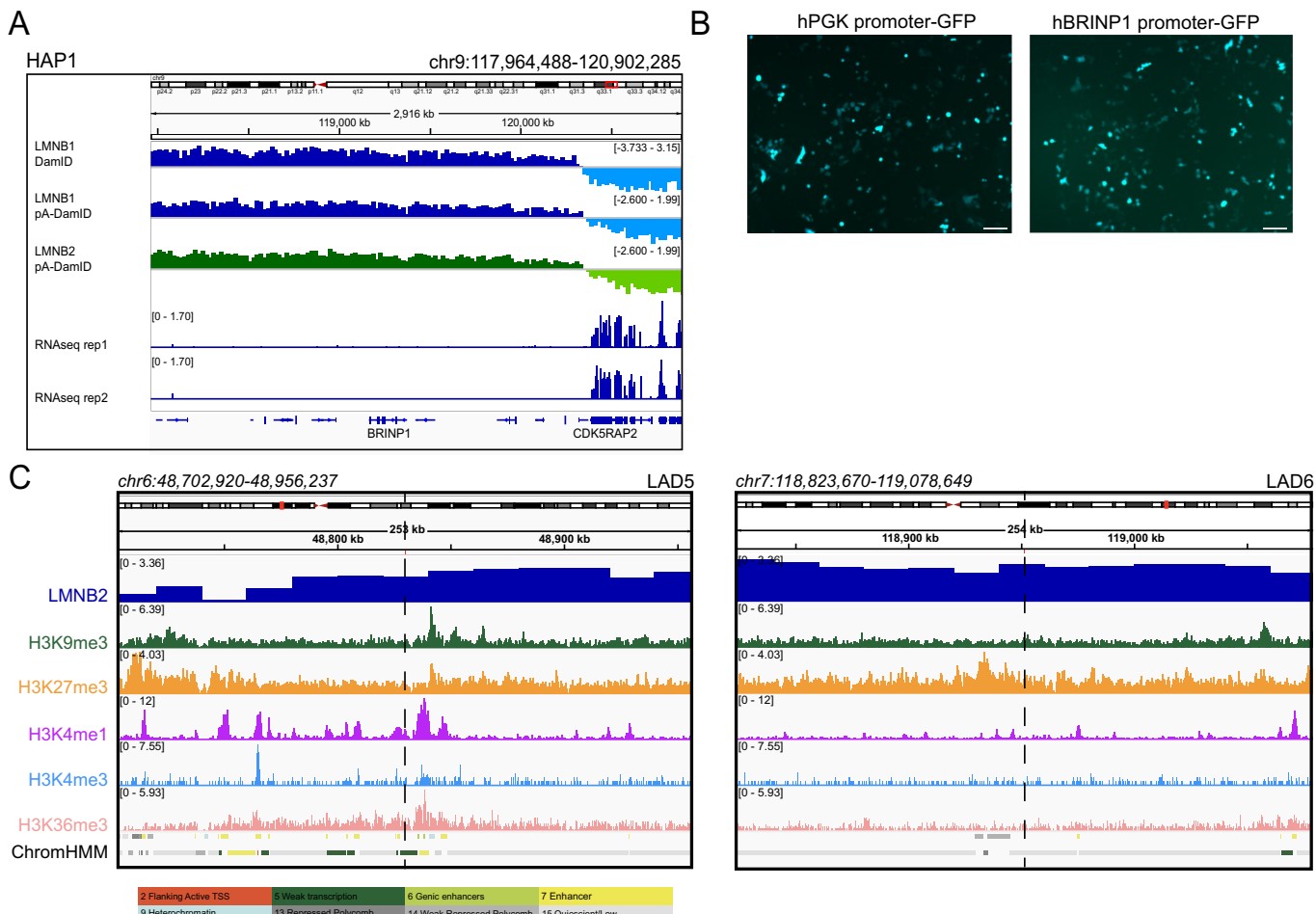

**Figure EV1.  Characterization of BRINP1 promoter and the chromatin environment around LAD reporters' integration sites in HAP1 cells.**

(**A**) Top tracks: IGV tracks showing DNA-NL contacts (generated by in vivo DamID, LMNB1 pA-DamID, and LMNB2 pA-DamID) of 3 Mb region surrounding the BRINP1 gene. Bottom tracks: RNA-seq levels for *BRINP1* and surrounding genes in the same 3 Mb genomic regions. (**B**) Transfection of HAP1 cells with *phPGK::GFP* or *pBRINP1::GFP* plasmids. *BRINP1* promoter is active in HAP1 cells when expressed in an episomal setting. Scale is at 20 μm. (**C**) DamID tracks for LMNB2 and ChIP-seq data for H3K9me3, H3K27me3, H3K4me1, H3K4me3, and H3K36me3 in HAP1 in a 250 kb window surrounding LAD5 and LAD6 reporter integrations (dashed lines). Data are from (Haarhuis et al, 2022). The bottom track shows annotations of major chromatin states according to ChromHMM (Ernst and Kellis, 2012); corresponding color key is shown in the bottom panel. Results are from at least two biological replicates.

## A  All Nuclear Envelope Proteins in LAD5 (Cheng et al, 2023)

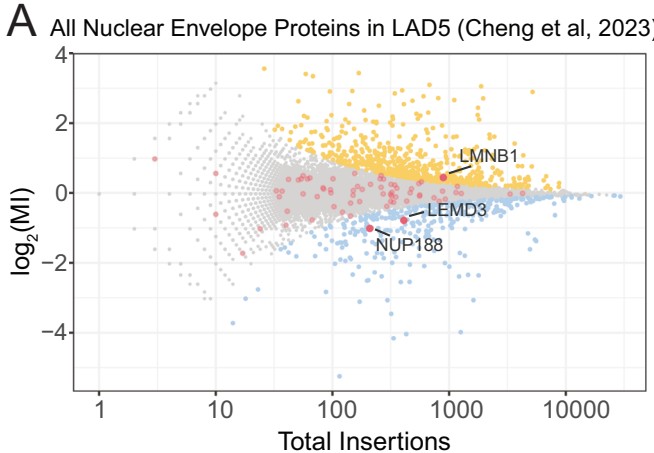

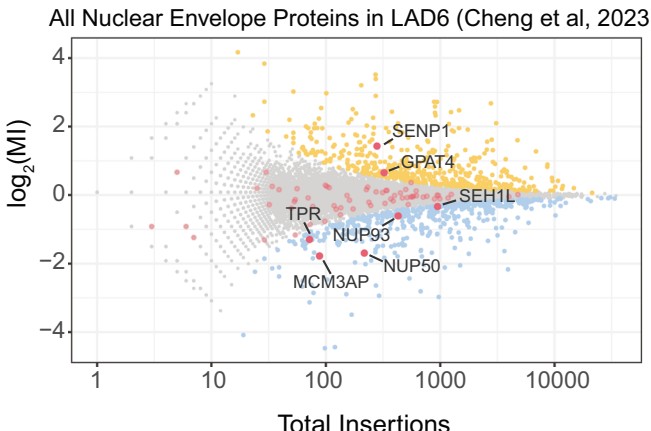

## B  Fitness score of Nuclear Envelope proteins

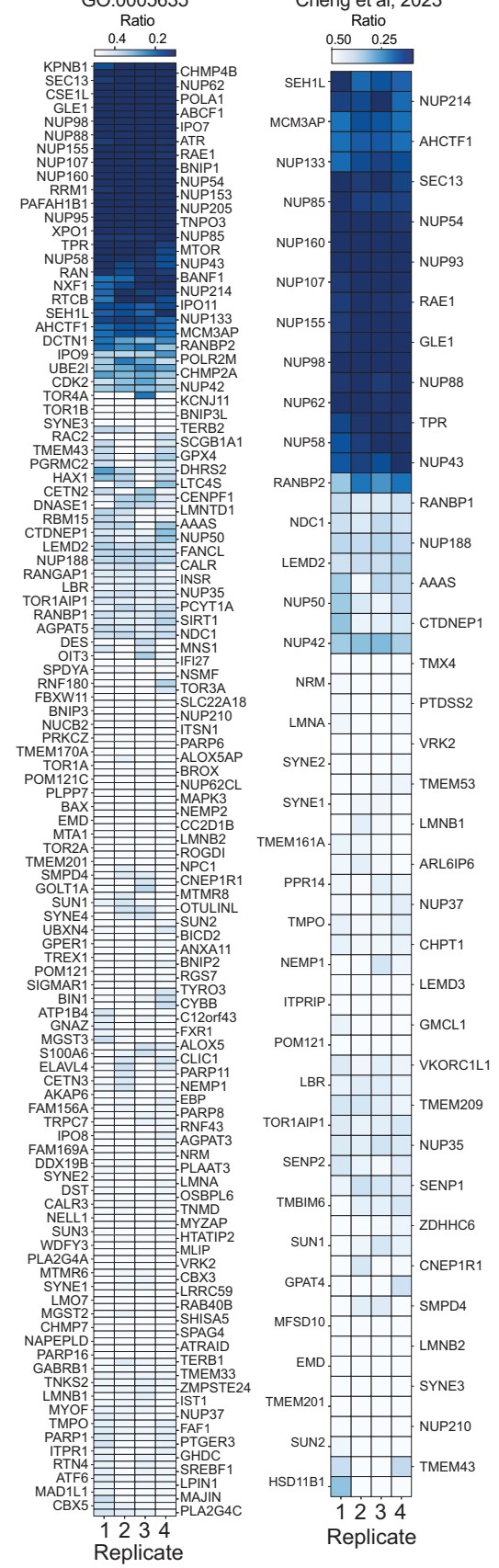

**Figure EV2.  Nuclear Lamina genes in the screen and their essentiality in HAP1 cells.**

(A) Biochemically identified NE proteins (Cheng et al, 2019) (red dots) highlighted in the screen fishtail plots for LAD5 (top) and LAD6 (bottom). The majority of these NE proteins are not significant hits in the screens. Names of the handful of significant screen hits are indicated in black. (B) Essentiality scores of proteins in GO category GO:0005635 (left panel) and of proteins biochemically identified as NE proteins (Cheng et al, 2019), (right panel). Heatmaps show the ratio of sense insertions to the total insertions in wild-type HAP1 cells across 4 independent replicates under untreated conditions. Data are from (Blomen et al, 2015). The scores represent the ratio of disruptive insertions (sense) to the total insertions (sense "disruptive" + antisense "non-disruptive") within the intronic regions of each gene. Genes crucial for cell viability will have fewer disruptive insertions as these cells are depleted, whereas cells with non-disruptive (antisense) insertions survive. As disruptive and non-disruptive integrations occur at similar frequencies, the ratio of insertions in the surviving population indicates whether a gene is important for cell fitness (Blomen et al, 2015). The lower the ratio (blue shading), the more important the gene is for HAP1 cell fitness. Results from 4 different biological replicates are shown separately.

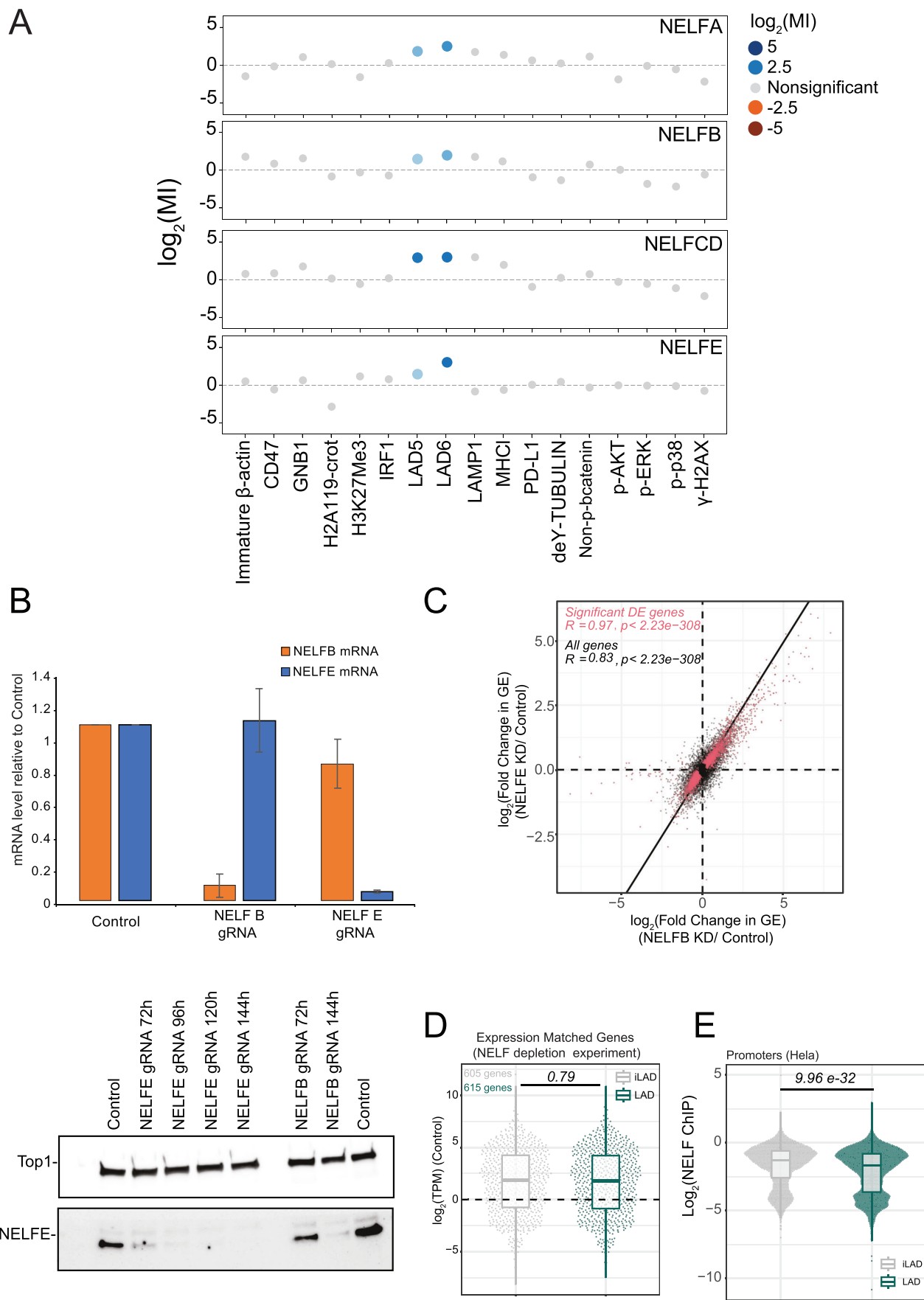

**Figure EV3.   NELF depletion in HAP1.**

(A) The mutational index (MI) for subunits from the NELF complex (NELFA, NEFLB, NELFCD, NELFE) was plotted for the current LAD5 and LAD6 screens and 15 additional FACS-based haploid screens previously conducted in HAP1 cells using the indicated readouts (blue, negative regulator; orange, positive regulator; gray, not significant) (Brockmann et al, 2017; Haahr et al, 2022; Jongsma et al, 2021; Logtenberg et al, 2019; Mazouzi et al, 2023; Mezzadra et al, 2017; Nieuwenhuis et al, 2017). (B) NELF depletion by CRISPRi in HAP1. Top panel: mRNA levels (measured by RT-qPCR) for *NELFE* and *NELFB* following CRISPRi depletion using *NELFE* and *NELFB* specific guide RNAs, 144 h after transduction. Data were first normalized on *GAPDH* mRNA and then on *NELFE/B* mRNA levels in control cells. The error bar represents standard deviation. Results were from three replicates. Bottom panel: detection of NELFE protein levels following CRISPRi depletion using *NELFE* and *NELFB* specific guide RNAs at different timepoints after transduction. DNA topoisomerase 1 (Top1) antibody was used as loading control. (C) Correlation between changes in gene expression following NELFB and NELFE knockdowns, for all (black) and significantly de-regulated genes (red). Results were from three replicates for NELFE depletion and two replicates for NELFB depletion. The black line is the diagonal. (D) Gene expression levels for expression-matched LAD and iLAD genes in the NELF depletion experiment. (E) NELFE levels at promoters of genes in LADs and expression-matched genes in iLADs in HeLa cells. ChIP-seq data are from (Beckedorff et al, 2020). Results are from two biological replicates. *P* value is according to Wilcoxon's test. For (D, E), the central line in the boxplots represents the median. The lower and upper hinges correspond to the first and third quartiles (the 25th and 75th percentiles). The upper and lower whiskers extends from the hinge to the largest or smallest values respectively no further than 1.5 * inter-quartile range. Outliers were removed only for visualization purposes. Source data are available online for this figure.

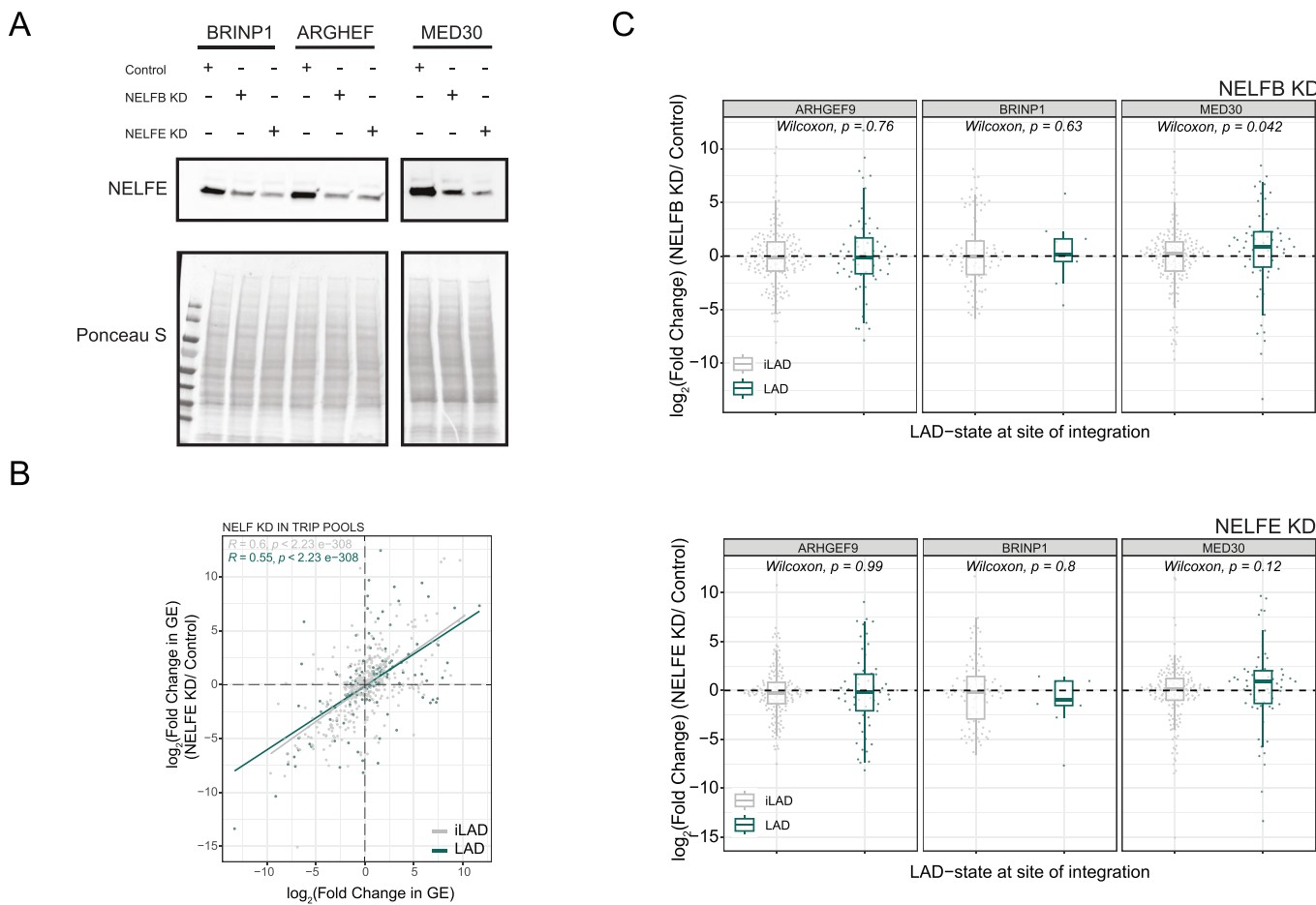

**Figure EV4. NELF depletion in K562 TRIP pools.**

(A–C) Multiplexed detection of the effect of depletion of NELFB and NELFE on the expression levels of reporter genes randomly integrated throughout the genome in K562 cell pools. Barcoded reporters were driven by promoters from the *ARHGEF9*, *BRINP1*, and *MED30* genes as indicated. Cell pools are from (Leemans et al, 2019). (A) Western blot of NELFE showing partial knockdown after siRNA-mediated depletion of NELFB or NELFE. (B) Changes in expression of the reporters for the three promoters throughout the genome correlate between NELFB and NELFE knockdowns. Reporters integrated in LADs are shown in green, reporters in iLADs are shown in gray. The gray and green lines represent a fitted linear model for iLAD and LAD integrations, respectively; Pearson correlation and *P* values are shown in the plots. (C) Changes in expression of each reporter (log₂ scale) after siRNA-mediated knock-down of NELFB (top panel) and NELFE (bottom panel), divided by location in either LADs or iLADs. Results are from three replicates for $P_{BRINP1}$ and two replicates for $P_{MED30}$ and $P_{ARHGEF9}$. *P*-values comparing the distributions in LADs and iLADs are according to Wilcoxon test. Source data are available online for this figure.

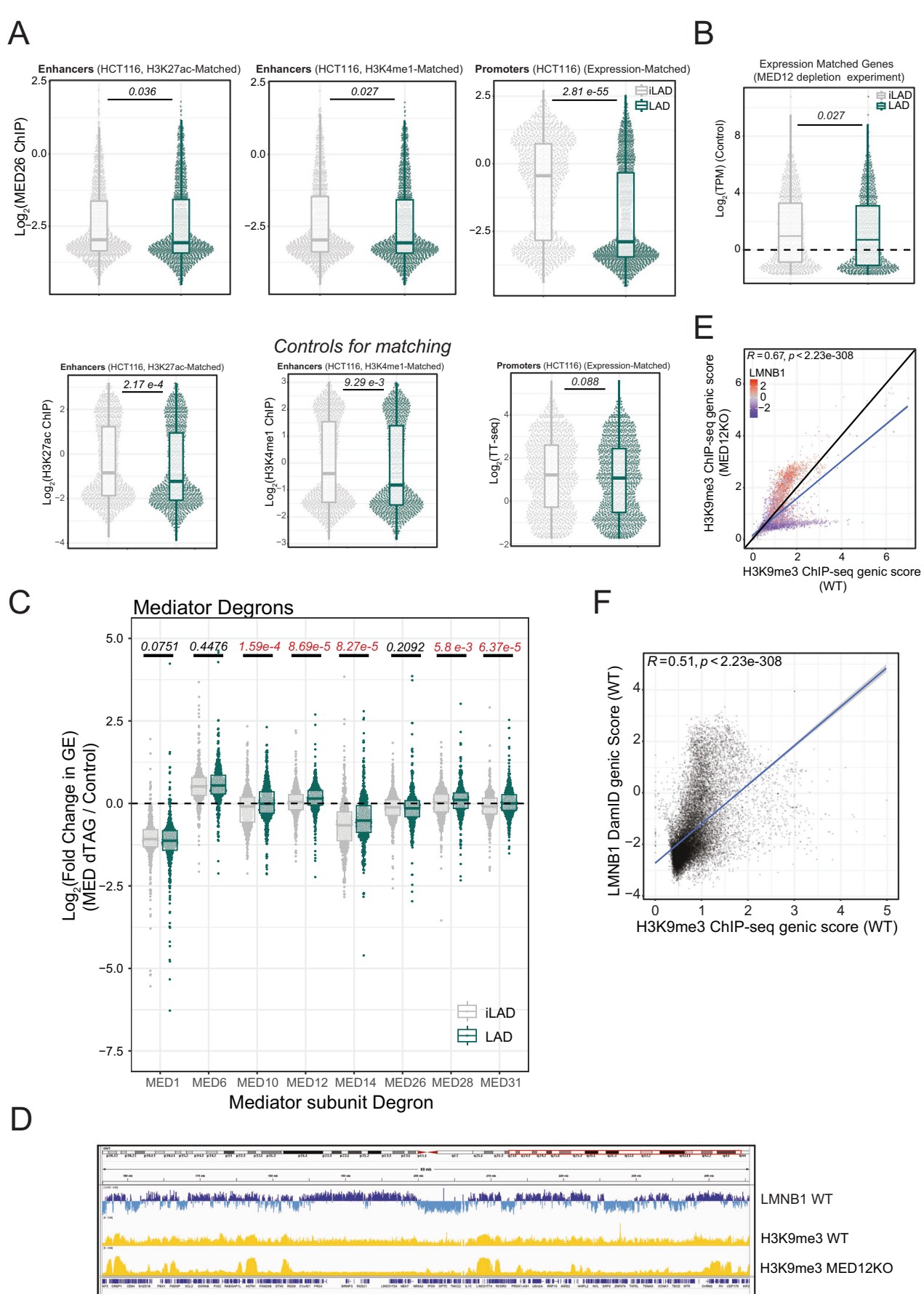

◄   **Figure EV5.   Characterization of Mediator complex in LADs.**

(**A**) Top panels: MED26 binding at promoters and enhancers in HCT116 cells. Promoters were matched for TT-seq level to compare LAD and iLAD genes with similar transcriptional activity. Enhancers were matched for H3K27ac or H3K4me1 levels to compare LAD and iLAD regulatory elements with similar enhancer activity. *P* values in are according to Wilcoxon's test. Bottom panels: plot showing the similar distributions of H3K27ac (left) or H3K4me1 (middle) for matched sets of enhancers; and TT-seq levels for matched sets of promoters (right) in LADs and iLADs.Data are from two biological replicates. (**B**) Gene expression levels for expression-matched LAD and iLAD genes in the MED12 depletion experiments. Data are from (Haarhuis et al, 2022) and results are from 6 biological replicates. (**C**) $\log_2$(fold change) in gene expression (GE) following acute depletion of Mediator subunits for expression-matched LAD and iLAD genes. Statistical significance was calculated with Wilcoxon test for comparison of median and significant *P* values are highlighted in red. Results are from three biological replicates (**D**) IGV genomic tracks for 89 Mb of Chromosome 1 showing LMNB1 DamID profile for HAP1 WT (blue) and H3K9me3 ChIP-seq scores for HAP1 WT and MED12 KO (yellow). The genomic tracks show increased compartmentalization of heterochromatin in LADs following MED12 knockout. (**E**) Correlation between H3K9me3 levels for genes in WT and MED12 knockout cell lines. Datapoints are colored by LMNB1 DamID score. (**F**) Correlation between genic LMNB1 DamID score in WT and H3K9me3 levels in WT cells. The blue line (**E**, **F**) represents a fitted linear model; Pearson correlation and *P* values are shown in the plots. Results (**E**, **F**) are from three biological replicates. Data are from (El Khattabi et al, 2019; Leemans et al, 2019; Lidschreiber et al, 2021; Schick et al, 2021; Haarhuis et al, 2022, 35136067). For (**A–C**), the central line in the boxplots represents the median. The lower and upper hinges correspond to the first and third quartiles (the 25th and 75th percentiles). The upper and lower whiskers extends from the hinge to the largest or smallest values respectively no further than 1.5 * inter-quartile range. Outliers were removed only for visualization purposes.

