## [Peer Review File · The EMBO Journal]

Chromatin protein complexes involved in gene repression in lamina-associated domains

Stefano Manzo, Abdelghani Mazouzi, Christ Leemans, Tom van Schaik, Nadia Neyazi, Marjon van Ruiten, Benjamin Rowland, Thijn Brummelkamp, and Bas van Steensel

Corresponding author: Bas van Steensel (b.v.steensel@nki.nl)

Review Timeline:

Submission Date:	26th Mar 24
Editorial Decision:	3rd May 24
Revision Received:	2nd Jul 24
Editorial Decision:	29th Jul 24
Revision Received:	5th Aug 24
Accepted:	7th Aug 24

Editor: Kelly Anderson

Transaction Report:

Dear Prof. van Steensel,

Thank you for submitting your manuscript for consideration by the EMBO Journal. It has now been seen by three referees whose comments are shown below.

Given the referees' positive recommendations, I would like to invite you to submit a revised version of the manuscript, addressing the comments of all three reviewers. I should add that it is EMBO Journal policy to allow only a single round of revision, and acceptance of your manuscript will therefore depend on the completeness of your responses in this revised version. It would be good to discuss your plan to address the referee concerns and I am available to do so in the coming weeks by zoom or by email.

Thank you for the opportunity to consider your work for publication. I look forward to your revision.

Yours sincerely,

Kelly M Anderson, PhD
Editor, The EMBO Journal
k.anderson@embojournal.org

We realize that it is difficult to revise to a specific deadline. In the interest of protecting the conceptual advance provided by the work, we recommend a revision within 3 months (1st Aug 2024). Please discuss the revision progress ahead of this time with the editor if you require more time to complete the revisions.

Referee #1:

The manuscript by Manzo et al tries to identify proteins that are impacting gene expression patterns within Lamina-associated domains (LADs). To accomplish this, they are using reporter genes that have been integrated into two LADs as well as an inter-LAD region in human HAP1 cells. Using this approach, the authors identify multiple proteins that are shared or specific for gene expression patterns at each of these integration sites. Notably, nuclear lamina proteins were not among their top hits, with NELF proteins and chromatin remodeling factors being their top hits. Overall, the methods for this manuscript appear strong and the main challenge is with generalization of the results from these two integration sites to the entire LAD genomic landscape.

Specifically, the approach that the authors take to compare signal between an integration site of a gene reporter in a LAD, using GFP, versus an iLAD site, using mCherry, and combining that with a gene trap system appears quite powerful.

My main comment has to do with that only two sites within LADs were chosen for interrogation, and the integration of their gene reporter into these two sites is different (i.e. it is a single reporter in the LAD5 location, and a tandem reporter in the LAD6 location). Furthermore, given the marked differences in their screen hits between these two sites, it is quite likely that if they were to include a third site, that they would get different results. The RNA-seq studies that they analyzed in parallel do help address this concern, but I do think that this limitation needs more discussion in the text. Specifically, their findings could be influenced by LAD-independent local chromatin features to a much higher degree than any LAD-specific features.

In addition to this major comment, I have several other minor comments:

The use of a haploid cell line will limit their screen, as genes that are conditionally lethal in HAP1 cells will not show up. Are nuclear lamina proteins conditionally lethal in HAP1 cells? This potential limitation of their screen should be mentioned.

Please expand on the statement on Page 5, lines 197-198 that "it is likely that many of these hits are due to indirect effects"

As mentioned above, differences between LAD5 and LAD6 could be driven by single versus tandem integration of reporter. This should be discussed.

The datapoints for the NELFB plot in Supplemental Figure 2 are not aligned appropriately. Please correct.

The legend for Figure 3D appears to be missing.

Page 13, lines 446-448, this sentence appears to be missing word "It was previously found that stable deletion of MED12 causes concentration of H3K9me3 in large genomic regions, while the rest of the genome becomes depleted of this heterochromatin mark."

Referee #2:

How gene expression is ensured in LADs has been a long-lasting pending question. In this manuscript, Manzo et al search for repressors of gene expression in LADs. They use a clever genome-wide genetic screen in the haploid HAP1 cell line, in which a reporter system has been inserted to monitor gene expression in two model LADs. Insertional mutagenesis and FACS using signals from the reporters enables them to identify putative LAD regulators. The screen leads to unexpectedly relatively few NE/NL proteins, but is enriched for chromatin regulators. The authors then go on to functionally validate three such regulators, such as NELF, BAF and subunits of the mediator complex, among which they more extensively tested MED12.

The results provide new insights on what controls the overall repressive state of LADs, and interestingly also point out that genes different LADs may be regulated (repressed) by distinct regulators. The work interestingly highlights a hitherto underappreciated functional heterogeneity of LADs in their repressive state. The data (hits) are also likely to constitute a useful resource for the field.

This is a clever and very well conducted study which is also very well written. I have no fundamental comments and recommend the manuscript to be published.

It would nonetheless be interesting to see a discussion on whether the authors believe the hits identified for LADs "5 and 6" would also apply to other LADs: would repressors of specific LADs also function in other LADs? It would also be interesting to get an idea of the relationship between the stochasticity of LADs - a view the authors seem to favor, and the specificity (?) of repressors enriched in specific LADs, such as those examined in this study.

Referee #3:

In this manuscript, Manzo and colleagues performed whole-genome genetic screens using LAD-integrated fluorescent reporters to identify such regulators of gene repression in the nuclear periphery. Perhaps surprisingly, they found that negative regulators of transcriptional elongation (e.g. NELF), as well as chromatin remodelers and Mediator complex components are the most impactful regulators, and therefore propose that "the fundamental regulatory steps of the transcription process and chromatin remodeling factors, rather than interaction with NL proteins, play a major role in the regulation of transcription within LADs".

The overall approach (unbiased screen in haploid cells), the structure of the manuscript (clearly written and with a very good flow of results and discussion), as well as the presentation of the data in the figures (intuitive and with enough detail in the figure legends) are of high quality. And, it is important to note, that - in hindsight - the concept of lamina-embedded genes being regulated via the transcriptional machinery, rather than primarily by NL components, is one that makes sense and appears to be well supported by the data (e.g. multiple such factors pop-up in the authors' screen). However, I would like to point out the following that I think need to be taken into consideration in order to fully consolidate the conclusions drawn in this manuscript (not in order of significance):

- The use of a single promoter; providing data from at least one more, different promoter would significantly strengthen the paper. I assume that performing a whole new screen is too much of an ask, but I could imagine simply generating a line with a second promoter construct in the LADs and then performing knockdowns with the top hits of the existing screen to validate (or not?) the results.
- The "modest overlap" of hits between the two LAD loci; this indeed points to very different machinery contributing to repression/regulation in the two loci, and this inevitably suggests that testing only two loci might provide low power of generalization of results. Since, again, redoing the screen by including more integration loci is too much of an ask, I wonder if the authors could compare the chromatin environment/nearby gene composition/etc of the two positions in an effort to identify what might be different between the two?
- NELF/DSIF components are top hits; by eliminating NELF/DSIF, the control mCherry construct would, if anything, not be affected, but can the authors really rule out the fact that this is an indirect effect on their LAD-embedded constructs? For example, loss of NELF/DSIF upregulates some TF/regulator/signaling gene(s) that suffice for the induction of gene expression in LADs. Perhaps these could be inferred from the CRISPRi knockdowns they perform next?
- Also, why CRISPRi for knocking-down NELF? It seems to me that 6 days is a long time and might also confer some compensatory events? And could the authors elaborate in more detail on the outcome of this knockdown test? The current text seems to brief for a validation experiment that can provide critical answers for the screen. By the way, I find panel 3B with the schematic of the NELF/DSIF/etc complex function to be unnecessary.
- BAF complex components; although the bimodal (stimulatory and repressive) effects by this chromatin remodelling complex are conceivable, I assume that the authors can again not rule out the possibility of indirect effects, like those for NELF/DSIF elimination discussed above. Moreover, could the presumed "slow response to BAF loss" not be also indicative of indirect effects for downregulation that requires much more time to manifest? Wouldn't this be the simplest explanation?
- Mediator components; the addition of ChIP data here is most welcome, but it highlights the lack of such data from previous figures. I think that providing direct evidence (e.g. ChIP-qPCR data) of NELF/DSIF, BAF subunits, MED subunits binding or not to the integrated constructs (as well as to the control one) would significantly strengthen the intimate relationship of the factors and the integrated promoter. Ideally, ChIP-seq (or equivalent) profiles of any of these subunits across LADs could be helpful. Also, if repression of elongation is such a prominent player (rather than instating of a fully repressive environment), one would expect to find RNAPII loaded to LAD promoters and to the constructs themselves. Is this the case or is this perhaps already known?
- Finally, the possibility of indirect effects from the screen should be commented upon in the Discussion of the manuscript, which is currently somewhat too concise for the rich implications of this dataset.

I typically disclose my identity to authors: A. Papantonis

Manzo et al, Point-by-point response

Black: reviewer comments

Blue: response.

Blue+bold: key modifications in the manuscript

Reviewer #1

The manuscript by Manzo et al tries to identify proteins that are impacting gene expression patterns within Lamina-associated domains (LADs). To accomplish this, they are using reporter genes that have been integrated into two LADs as well as an inter-LAD region in human HAP1 cells. Using this approach, the authors identify multiple proteins that are shared or specific for gene expression patterns at each of these integration sites. Notably, nuclear lamina proteins were not among their top hits, with NELF proteins and chromatin remodeling factors being their top hits. Overall, the methods for this manuscript appear strong and the main challenge is with generalization of the results from these two integration sites to the entire LAD genomic landscape.

Specifically, the approach that the authors take to compare signal between an integration site of a gene reporter in a LAD, using GFP, versus an iLAD site, using mCherry, and combining that with a gene trap system appears quite powerful.

My main comment has to do with that only two sites within LADs were chosen for interrogation, and the integration of their gene reporter into these two sites is different (i.e. it is a single reporter in the LAD5 location, and a tandem reporter in the LAD6 location). Furthermore, given the marked differences in their screen hits between these two sites, it is quite likely that if they were to include a third site, that they would get different results. The RNA-seq studies that they analyzed in parallel do help address this concern, but I do think that this limitation needs more discussion in the text. Specifically, their findings could be influenced by LAD-independent local chromatin features to a much higher degree than any LAD-specific features.

Indeed it is likely that performing the screen on a third LAD reporter would have yielded partially different results, although the results for several factors (for example NELF and associated proteins) appeared very robust between the two LADs. **We have modified both our Results (lines ##162-167; ##272; ##349-##369) and Discussion (final paragraph) to discuss this limitation and highlight the importance of local chromatin context in the regulation of gene expression in LADs.**

Please note that these screens were quite laborious. Aside from the establishment and characterization of the reporter cell lines, we needed to process 10 billion cells through sorting for both cell lines (LAD5 and LAD6), which required about 80 x T175 flasks. This process took 5 days of sorting for roughly 10 hours per day. In addition, each screen required ~800 million sequence reads. **We now clarify these technical aspects in the Methods.**

The use of a haploid cell line will limit their screen, as genes that are conditionally lethal in HAP1 cells will not show up. Are nuclear lamina proteins conditionally lethal in HAP1 cells? This potential limitation of their screen should be mentioned.

Good point. We now analysed previous measurements of "essentiality" in HAP1 cells (Blomen et al. 2015). About 25-30% of the NE/NL proteins affect cellular fitness substantially, and thus may have been missed in the screen. However, the vast majority of these proteins are nuclear pore or nuclear transport proteins. **Results of this analysis are reported in lines ##240-##251 of the Results, and Fig EV2B.**

Please expand on the statement on Page 5, lines 197-198 that "it is likely that many of these hits are due to indirect effects".

We modified this sentence by providing more concrete examples of indirect effects. Now lines ##206-##207.

As mentioned above, differences between LAD5 and LAD6 could be driven by single versus tandem integration of reporter. This should be discussed.

We have added "Limitations of the study" section at the end of the Discussion, that includes this aspect.

The datapoints for the NELFB plot in Supplemental Figure 2 are not aligned appropriately. Please correct.

We appreciate the comment, but there is no NELFB plot in Sup Fig2. Perhaps the reviewer meant Supp Fig 5B (FIG EV3C in the revised manuscript), and noticed that the fitted line does not align with the datapoints? **We clarified that this line is the diagonal, not a fitted line (now line ##957).**

The legend for Figure 3D appears to be missing.

This part of the figure legend was simply masked by the textbox border - our apologies. We have corrected it.

Page 13, lines 446-448, this sentence appears to be missing word "It was previously found that stable deletion of MED12 causes concentration of H3K9me3 in large genomic regions, while the rest of the genome becomes depleted of this heterochromatin mark."

We have modified this sentence (now lines ##464-##466).

Reviewer #2

How gene expression is ensured in LADs has been a long-lasting pending question. In this manuscript, Manzo et al search for repressors of gene expression in LADs. They use a clever genome-wide genetic screen in the haploid HAP1 cell line, in which a reporter system has been inserted to monitor gene expression in two model LADs. Insertional mutagenesis and FACS using signals from the reporters enables them to identify putative LAD regulators. The screen leads to unexpectedly relatively few NE/NL proteins, but is enriched for chromatin regulators. The authors then go on to functionally validate three such regulators, such as NELF, BAF and subunits of the mediator complex, among which they more extensively tested MED12.

The results provide new insights on what controls the overall repressive state of LADs, and interestingly also point out that genes different LADs may be regulated (repressed) by distinct regulators. The work interestingly highlights a hitherto underappreciated functional heterogeneity of LADs in their repressive state. The data (hits) are also likely to constitute a

useful resource for the field.

This is a clever and very well conducted study which is also very well written. I have no fundamental comments and recommend the manuscript to be published.

It would nonetheless be interesting to see a discussion on whether the authors believe the hits identified for LADs "5 and 6" would also apply to other LADs: would repressors of specific LADs also function in other LADs? It would also be interesting to get an idea of the relationship between the stochasticity of LADs - a view the authors seem to favor, and the specificity (?) of repressors enriched in specific LADs, such as those examined in this study.

In the final paragraph of the Discussion, we now elaborate more on impact of the local chromatin makeup of LADs. While we indeed think that LAD-NL contacts show variable degrees of stochasticity, this seems less relevant here, because our screens identified very few NL components. Contacts with the NL may thus be less important than local chromatin regulation, as we highlighted in this manuscript.

Reviewer #3

In this manuscript, Manzo and colleagues performed whole-genome genetic screens using LAD-integrated fluorescent reporters to identify such regulators of gene repression in the nuclear periphery. Perhaps surprisingly, they found that negative regulators of transcriptional elongation (e.g. NELF), as well as chromatin remodelers and Mediator complex components are the most impactful regulators, and therefore propose that "the fundamental regulatory steps of the transcription process and chromatin remodeling factors, rather than interaction with NL proteins, play a major role in the regulation of transcription within LADs".

The overall approach (unbiased screen in haploid cells), the structure of the manuscript (clearly written and with a very good flow of results and discussion), as well as the presentation of the data in the figures (intuitive and with enough detail in the figure legends) are of high quality. And, it is important to note, that - in hindsight - the concept of lamina-embedded genes being regulated via the transcriptional machinery, rather than primarily by NL components, is one that makes sense and appears to be well supported by the data (e.g. multiple such factors pop-up in the authors' screen). However, I would like to point out the following that I think need to be taken into consideration in order to fully consolidate the conclusions drawn in this manuscript (not in order of significance):

- The use of a single promoter; providing data from at least one more, different promoter would significantly strengthen the paper. I assume that performing a whole new screen is too much of an ask, but I could imagine simply generating a line with a second promoter construct in the LADs and then performing knockdowns with the top hits of the existing screen to validate (or not?) the results.

Indeed, repeating the screens with additional reporters is beyond the scope of this paper, as the screen design is technically very challenging. Aside from the establishment and characterisation of the reporter cell lines, we needed to process 10 billion cells through sorting for both cell lines (LAD5 and LAD6), which required about 80 x T175

flasks. This process took 5 days of sorting for roughly 10 hours per day. In addition, each screen required nearly 1 billion sequence reads. **We now clarify these technical aspects in the Methods.**

To partially address the reviewer's point, we now used our TRIP method to measure the impact of NELF depletion on reporters integrated throughout the genome, including in LADs. We did this for three different promoters. The results show that one promoter is more sensitive than the others and shows preferential upregulation in LAD following NELF loss. An additional take-home message from these new experiments is that the local chromatin context within LADs has strong impact on the responsiveness of promoters to NELF. **These new results are presented in lines ##348-## Fig EV4A-C 369 and Figures Fig EV4A-C.**

- The "modest overlap" of hits between the two LAD loci; this indeed points to very different machinery contributing to repression/regulation in the two loci, and this inevitably suggests that testing only two loci might provide low power of generalization of results. Since, again, redoing the screen by including more integration loci is too much of an ask, I wonder if the authors could compare the chromatin environment/nearby gene composition/etc of the two positions in an effort to identify what might be different between the two?

We now analysed the chromatin state of our two integration sites (lines ##162-##167 and new Fig EV1C). The local chromatin context of the two LADs is clearly different. We highlight these differences as possible explanation for the opposite screen results for BAF (line ##381) and Mediator (lines ##434) complexes. We also emphasize the importance of local chromatin context in the Discussion (lines ##550-##552; ##576-##600)

- NELF/DSIF components are top hits; by eliminating NELF/DISF, the control mCherry construct would, if anything, not be affected, but can the authors really rule out the fact that this is an indirect effect on their LAD-embedded constructs? For example, loss of NELF/DISF upregulates some TF/regulator/signaling gene(s) that suffice for the induction of gene expression in LADs. Perhaps these could be inferred from the CRISPRi knockdowns they perform next?

This is a fair point, although we find it striking that the screen not only identified NELF, but also various other proteins implicated in regulation of promoter-proximal pausing. It seems somewhat unlikely that all these proteins would act on the LAD reporters via the same indirect mechanism, but it cannot be fully ruled out. We believe that it would be very challenging to discriminate between direct and indirect effects from our RNA-seq data. To address this and other reviewers' comments **we have included a "limitation of the study" paragraph at the end of the Discussion that addresses the possibility that our screens and long-term depletions could catch indirect effects.**

- Also, why CRISPRi for knocking-down NELF? It seems to me that 6 days is a long time and might also confer some compensatory events? And could the authors elaborate in more detail on the outcome of this knockdown test? The current text seems to brief for a validation experiment that can provide critical answers for the screen.

We attempted to make inducible NELFE and NELFB degron HAP1 cell clones, but we did not manage to obtain convincing depletion. We therefore resorted to CRISPRi, which was more effective but indeed slower.

We devoted an entire section of text (lines **##323-##347**) and Figures 3B-C and **Fig EV3D-E** to the analysis of these knockdown experiments. We are not sure what the reviewer means by "brief" – please let us know what is specifically missing, then we will be happy to elaborate. **We now also added an analysis of NELF ChIP-seq data (new Figure FigEV3E; Results lines ##343-346).**

- By the way, I find panel 3B with the schematic of the NELF/DISF/etc complex function to be unnecessary.

We have removed this panel.

- BAF complex components; although the bimodal (stimulatory and repressive) effects by this chromatin remodelling complex are conceivable, I assume that the authors can again not rule out the possibility of indirect effects, like those for NELF/DSIF elimination discussed above. Moreover, could the presumed "slow response to BAF loss" not be also indicative of indirect effects for downregulation that requires much more time to manifest? Wouldnt this be the simplest explanation?

We agree. **We have added on line ##440: "This could indicate that the effect of BAF on LAD genes is indirect."**

- Mediator components; the addition of ChIP data here is most welcome, but it highlights the lack of such data from previous figures. I think that providing direct evidence (e.g. ChIP-qPCR data) of NELF/DISF, BAF subunits, MED subunits binding or not to the integrated constructs (as well as to the control one) would significantly strengthen the intimate relationship of the factors and the integrated promoter. Ideally, ChIP-seq (or equivalent) profiles of any of these subunits across LADs could be helpful.

We added a further analysis of available NELFE ChIP-seq data in HeLa cells. To enable this analysis, we generated new NL interaction maps in HeLa. **We now present these results in lines 356-360## and FigEV3E.**

Generation of reliable ChIP data for Mediator is very challenging; this has been documented in detail (El Khattabi et al 2019, PMID:31402173). Nevertheless, one trustworthy dataset (for MED26) provided an interesting piece of the puzzle: we found that this subunit typically only binds to enhancers, but rarely to promoters in LADs (unlike in iLADs, where it frequently interacts with promoters). To enable this analysis we generated new NL interaction maps in the matching cell type (HTC116). **We present these new results on lines ##430-##436 and Fig EV5A**, and suggest that the activation of the reporter in LAD5 by Mediator is explained by its proximity to enhancer-like elements, which are not present in LAD6.

For BAF we have consulted experts in the field, who discouraged us from venturing into ChIP-seq of BAF subunits. Apparently, ChIP of the BAF complex is extremely unreliable, and may suffer from strong biases toward hyperaccessible chromatin, making interpretation of results even more challenging.

- Also, if repression of elongation is such a prominent player (rather than instating of a fully repressive environment), one would expect to find RNAPII loaded to LAD promoters and to the constructs themselves. Is this the case or is this perhaps already known?

This was indeed known, it matches two previous reports; **we mention this in the Discussion (lines ##524-##526).**

- Finally, the possibility of indirect effects from the screen should be commented upon in the Discussion of the manuscript, which is currently somewhat too concise for the rich implications of this dataset.

We have further emphasized this throughout the manuscript (lines 412-413##; ##461-), in particular in a new final paragraph that discusses the limitations of the study more broadly (lines 615-636##). We have also expanded our discussion proposing mechanistic speculations for BAF and Mediator.

Dear Prof. van Steensel,

Congratulations on a great revision! Overall, the referees have been positive however one referee had one remaining concern that we ask you to (non-experimentally) address in a revised version. When you submit your revised version, please also take care of the following editorial items and add this also to your point-by-point response:

1. Please include the following funding information onto our eJP online system: 838555, Next generation EU-MUR MSCA Young Researcher, institutional grant of the Dutch Cancer Society and of the Dutch Ministry of Health, Welfare and Sport, the Dutch Cancer Society
2. Please reduce the number of keywords to 5.
3. Please remove the author contribution section from main manuscript.
4. Please ensure the following figure referrals are correct in the main manuscript: Supplementary Figure 1C, D; Supplementary Figure 1C, E; Supplementary Figure 2A
5. Table S1 looks like a Dataset, if so it should be uploaded as such and the file name, title and legend should be corrected to Dataset EV1; the legend should be removed from the ms file and provided in the Excel sheet (e.g. as a separate sheet/tab).
6. Please add page numbers in the table of contents to the appendix file.
7. We require the publication of source data, particularly for electrophoretic gels and blots and graphs, with the aim of making primary data more accessible and transparent to the reader. It would be great if you could provide me with a PDF file per figure that contains the original, uncropped and unprocessed scans of all or key gels used in the figure or for graphs, an Excel spreadsheet with the original data used to generate the graphs. The PDF files should be labeled with the appropriate figure/panel number, and should have molecular weight marker; further annotation could be useful but is not essential. The PDF files will be published online with the article as supplementary "Source Data" files.
8. We include a synopsis of the paper (see <http://emboj.embopress.org/>). Please provide me with a general summary statement and 3-5 bullet points that capture the key findings of the paper.
9. We also need a summary figure for the synopsis. The size should be 550 wide by 200-440 high (pixels). You can also use something from the figures if that is easier.
10. Please upload the R&T table separately using the latest template.
11. Please remove the highlights section from the main manuscript.
12. Please rename the Methods and Protocols sections to "Methods".
13. Please update the section order to: Title page - Abstract & Keywords - Introduction - Results - Discussion - Methods - Data Availability - Acknowledgments - Disclosure Statement & Competing Interests - References - Figure Legends - (Main Tables with legends) - Expanded View Figure Legends.
14. In Figure EV4A, ARGHEF and MED30 appear to be from different blots. This should be indicated clearly with a white box separating the blots from the other series of blots.
15. In the data availability statement, please provide specific URLs for GSE261955, PRJNA1089502, PRJNA1087589 datasets.
16. Please note that the exact p values are not provided in the legends of figures 2c; EV 3c, e; EV 4b; EV 5a, e-f.
17. Please indicate the statistical test used for data analysis in the legends of figures 4c-d, f; EV 4b.
18. Please note that the box plots need to be defined in terms of minima, maxima, centre, bounds of box and whiskers, and percentile in the legends of figures 5e, g; EV 3d-e; EV 5a-c.
19. Please note that information related to n is missing in the legends of figures 5e, g; EV 3b; EV 5b.
20. Please note that the error bars are not defined in the legend of figure EV 3b.
21. Please note that scale bar and its definition are missing for figure EV 1b.

Thank you for the opportunity to consider your work for publication, I look forward to you revision.

Yours sincerely,

Kelly M Anderson, PhD
Editor, The EMBO Journal
k.anderson@embojournal.org

Referee #1:

Per my prior comment, the datapoints for the NELFB plot in Figure EV3A are not aligned appropriately (looks like one is missing). Sorry for saying the wrong figure previously. Please correct.

Otherwise, the authors have addressed all of my concerns.

Referee #2:

The authors have satisfactorily addressed my comments, and it seems, have done a thorough job addressing the other reviewers' comments as well. I have no further comments.

Referee #3:

The authors have thoroughly addressed all my remarks from the first revision round, and I particularly appreciate the addition of the "Limitations of the study" section in the Discussion that nicely sums up what we learn from this work, and what might be further needed. This was already a solid piece of work, but the revised manuscript is much better (analysis of more data is very helpful) and certainly mature enough to be published in its current form.

A. Papantonis

The authors addressed the minor editorial issues.

Dear Prof. van Steensel,

Congratulations on an excellent manuscript, I am pleased to inform you that your manuscript has been accepted for publication in The EMBO Journal. Thank you for your comprehensive response to the referee concerns and for providing detailed source data. It has been a pleasure to work with you to get this to the acceptance stage.

I will begin the final checks on your manuscript before submitting to the publisher next week. Once at the publisher, it will take about 3 weeks for your manuscript to be published online. As a reminder, the entire review process, including referee concerns and your point-by-point response, will be available to readers.

I will be in touch throughout the final editorial process until publication. In the meantime, I hope you find time to celebrate!

Warm wishes,
Kelly

Kelly M Anderson, PhD
Editor, The EMBO Journal
k.anderson@embojournal.org
